



# Effects of heterogeneous reactions on global tropospheric chemistry

Phuc T. M. Ha[1], Fumikazu Taketani[2], Yugo Kanaya[2], Ryoki Matsuda[1], Kengo Sudo[1,2]

[1] Graduate School of Environmental Studies, Nagoya University, Nagoya, 464-8601, Japan
[2] Research Institute for Global Change, JAMSTEC, Yokohama, 236-0001, Japan

*Correspondence to*: Phuc T. M. Ha (ha.thi.minh.phuc@f.mbox.nagoya-u.ac.jp)

**Abstract.** This study uses a chemistry-climate model CHASER (MIROC) to explore the roles of heterogeneous reactions (HRs) in global tropospheric chemistry. Three distinct HRs of $N_2O_5$, $HO_2$, and $RO_2$ are considered for surfaces of aerosols and cloud particles. The model simulation is verified with EANET and EMEP stationary observations, R/V MIRAI ship-based

data, ATOM1 aircraft measurements, satellite observations by OMI, ISCCP, and CALIPSO-GOCCP, and reanalysis data JRA55. The heterogeneous chemistry facilitates improvement of model performance with respect to observations for $NO_2$, OH, CO, and $O_3$, especially in the lower troposphere. The calculated effects of heterogeneous reactions cause marked changes in global abundances of $O_3$ (-3%), $NO_x$ (-2.2%), CO (+3.3%), and global mean $CH_4$ lifetime (+5.9%). These global effects were contributed mostly by $N_2O_5$ uptake onto aerosols in the middle troposphere. At the surface, $HO_2$ uptake gives the largest

contributions, with a particularly significant effect in the North Pacific region (-24% $O_3$, +68% $NO_x$, +8% CO, and -70% OH), mainly attributable to its uptake onto clouds. The $RO_2$ reaction has a small contribution, but its global-mean negative effect on $O_3$ is not negligible. In general, the uptakes onto ice crystals and cloud droplets that occur mainly by $HO_2$ and $RO_2$ radicals cause smaller global effects than the aerosol-uptake effects by $N_2O_5$ radicals (+1.34% $CH_4$ lifetime, +1.71% $NO_x$, -0.56% $O_3$, +0.63% CO abundances). Nonlinear responses of tropospheric $O_3$, $NO_x$, and OH to the $N_2O_5$ and $HO_2$ uptakes are found in the

same modelling framework of this study ($R > 0.93$). Although all HRs showed negative tendencies for OH and $O_3$ levels, the effects of $HR(HO_2)$ on the tropospheric abundance of $O_3$ showed a small increment with an increasing loss rate. However, this positive tendency turns to reduction at higher rates (>5 times). Our results demonstrate that the HRs affect not only polluted areas but also remote areas such as the mid-latitude sea boundary layer and upper troposphere. Furthermore, $HR(HO_2)$ can bring challenges to pollution reduction efforts because it causes opposite effects between $NO_x$ (increase) and surface $O_3$

(decrease).

## 1 Introduction

Heterogeneous reactions (HRs) on the surfaces of atmospheric aerosols and cloud droplets are regarded as playing crucial roles in atmospheric chemistry. They affect ozone ($O_3$) concentrations in various pathways via the cycle of odd hydrogens ($HO_x$) and nitrogen oxides ($NO_x$) (Jacob, 2000). Tropospheric ozone, an important greenhouse gas, causes damage to human health,





crop, and ecosystem productivity (Monks et al., 2015). Although tropospheric $O_3$ was recognized as a critical oxidant species, its global distribution has not been adequately captured to date because of the limited number of observations. Whereas many sites in the heavily polluted regions of eastern Asia show ozone increases since 2000 (Liu and Wang, 2020), many sites in other regions show decreases (Gaudel et al., 2018). Moreover, $O_3$ responds to changes of multiple pollutants such as $NO_x$ and VOCs in different ways, which challenge the local pollutant control policy. For instance, since the Chinese government

released the Air Pollution Prevention and Control Action Plan in 2010 (Zheng et al., 2018), the targets of $SO_2$, $NO_x$, and particulate matter (PM) decreased drastically, but urban ozone pollution has been worsening (Liu and Wang, 2020). Indeed, the $O_3$ responses are controlled by several mechanisms including heterogeneous effects of $HO_2$ and $N_2O_5$ onto aerosols (Kanaya et al., 2009; Taketani et al., 2012; Li et al., 2019; Liu and Wang, 2020).

Stationary observations and laboratory experiments are important for enhancing the understanding of the tropospheric

chemistry of $O_3$ and other essential components ($NO_x$, $HO_x$). However, direct observation of vertical $O_3$ distribution including upper tropospheric $O_3$ was not available before 1970. It has been deployed only at limited sites of the globe. Global atmospheric modelling is a useful method to reanalyze or to forecast the past and future changes in $O_3$ and their effects on human health and climate. To serve this task, atmospheric models use both laboratory and observational data to help achieve accurate simulations of $O_3$ and its precursors ($HO_x$, $NO_x$, hydrocarbons). To date, many modelling studies have suggested that

heterogeneous chemistry be included in a standard model for tropospheric chemistry (Jacob, 2000; MacIntyre and Evans, 2010, 2011; de Reus et al., 2005).

One fundamentally important HR in the troposphere is the uptake of $N_2O_5$ onto aqueous aerosols, known as a removal pathway for $NO_x$ at night (Platt et al., 1984). Actually, $NO_x$ plays crucially important roles in the troposphere because it controls the cycle of $HO_x$ and the production rate of tropospheric $O_3$ (Logan et al., 1981; Riemer et al., 2003). The morning

photochemistry can be affected by $NO_3$ and $N_2O_5$, which are important nocturnal oxidants. Since the early 1980s, the role of urban $NO_x$ chemistry in Los Angeles pollution (National Research Council, 1991) has been acknowledged, but the proclamation of nighttime radicals remained sparse. It was only recognized in the past decade that $N_2O_5$ radical chemistry could have a much more perceptible effect stemming from reasons counting a refined understanding of heterogeneous processes occurring at night (Brown and Stutz, 2012). The HR of $N_2O_5$ was revealed under different meteorological conditions

in the US, Europe, and China (photosmog, high relative humidity (RH), or seasonal variation) for particles of various types: ice, aqueous aerosols with organic-coating, urban aerosols, dust, and soot (Apodaca et al., 2008; Lowe et al., 2015; Qu et al., 2019; Riemer et al., 2003, 2009; Wang et al., 2018; Wang et al., 2017; Xia et al., 2019). The uptake of $N_2O_5$ can markedly enhance nitrate concentration in nocturnal chemistry or $PM_{2.5}$ explosive growth events in summer, decrease $NO_x$, and either increase or decrease $O_3$ concentrations in different $NO_x$ conditions (Dentener and Crutzen, 1993; Qu et al., 2019; Riemer et

al., 2003; Wang et al., 2017). Even during daytime, $N_2O_5$ in the marine boundary layers can enhance the $NO_x$ to $HNO_3$ conversion, and chemical destruction of $O_3$ (Osthoff et al., 2006). The 10–20 ppbv reduction of $O_3$ because of $N_2O_5$ uptake in the polluted regions of China has also been reported (Li et al., 2018). At mid- to high latitudes, $N_2O_5$ uptakes on sulfate aerosols





could engender 80% and 10% $NO_x$ reduction, respectively, in winter and summer, leading to approximate 10% reduction of $O_3$ in both seasons (Li et al., 2018).

Another vital process taking place on particles is the HRs of peroxy radicals ($HO_2$ and $RO_2$). Peroxy radicals are the primary chain carriers driving $O_3$ production in the troposphere. Moreover, it can drive the hydrocarbons and $NO_x$ concentration which are important for nocturnal radical chemistry (Geyer et al., 2003; Richard, 2000; Salisbury et al., 2001). In the past, the HR($HO_2$) effects have been well considered in the laboratory (Macintyre and Evans, 2011) and field observations (Kanaya et al., 2001, 2002a, 2002b, 2003, 2007; Taketani et al., 2012), but many technical problems (e.g.,

detecting $HO_2$) have created difficulties that challenge its reported importance in the troposphere, as asserted from recent studies (Liao and Seinfeld, 2005; Martin et al., 2003; Tie et al., 2001). More recently, global modelling reports have described that the inclusion of $HO_2$ uptake can affect atmospheric constituents strongly by the increment in tropospheric abundances for carbon monoxide (CO) and other trace gases because of reduced oxidation capacity (Lin et al., 2012; Macintyre and Evans, 2011). The $HO_x$ loss on aerosols can reduce $O_3$ concentrations by up to 33% in remote areas and up to 10% in a smog episode

(Saathoff et al., 2001; Taketani et al., 2012). The $HO_x$ loss on sea-salt, sulfate, and organic carbon in various environments can decrease respectively $HO_2$ levels by 6–13%, 10–40%, and 40–70% (Martin et al., 2003; Taketani et al., 2008, 2009; Tie et al., 2001). For $RO_2$ with a typical representative of $CH_3CO.O_2$ (peroxyacetyl radical, PA), it plays a big role in the long-range transport of pollution (VOC, $NO_x$) (Richard, 2000; Villalta et al., 1996). It can bring $NO_x$ from polluted domains as PAN to remote regions in the ocean and higher altitudes (Qin et al., 2018; Richard, 2000). The concentrations of $HO_2$ and $RO_2$ at

nighttime in the marine boundary layer were measured and confirmed (Geyer et al., 2003; Salisbury et al., 2001). Moreover, some evidence suggests uptake of $HO_2$ and PA on clouds, aqueous aerosols, and other surfaces in high humidity conditions, although the mechanism is uncertain (Geyer et al., 2003; Jacob, 2000; Kanaya et al., 2002b; Liao and Seinfeld, 2005; Lin et al., 2012; Richard, 2000; Salisbury et al., 2001). The predominance of peroxy uptake to clouds results from the ubiquitous existence and larger SAD maxima of cloud droplets in the atmosphere. Indeed, aqueous-phase chemistry might represent an

important sink for $O_3$ (Lelieveld and Crutzen, 1990). Also, PA loss on aqueous particles can mediate the loss of PAN ($CH_3CO.O_2NO$) in fog (Villalta et al., 1996). Some modelling studies indicate that $HO_x$ loss (including $HO_2$ loss) on aqueous aerosols causes about 2% reduction, 7% and 0.5% increments, respectively, in the annual mean global burden of OH, CO, and $O_3$ (Huijnen et al., 2014). However, in a coastal environment in the Northern Hemisphere it increases 15% OH and reduces 30% $HO_2$ (Sommariva et al., 2006; Thornton et al., 2008).

Although the contributions of each uptake category to tropospheric chemistry differ and must be considered both separately and as a whole, few studies have provided a global overview of heterogeneous chemistry the comprehensively examines the uptakes of $N_2O_5$, $HO_2$, and $RO_2$ on widely various particles. For instance, both uptakes of $N_2O_5$ and $HO_2$ tend to reduce $O_3$ in particular environments (Li et al., 2018; Saathoff et al., 2001; Taketani et al., 2012), but the $HO_2$ loss on clouds can increase the tropospheric $O_3$ burden (Huijnen et al., 2014). The latter trend is not widely suggested yet because the cloud

chemistry is still neglected in many $O_3$ models (Stadtler et al., 2018; Thornton et al., 2008). The predominant effects of $HO_2$ uptake on aerosols compared to the effect by $N_2O_5$ were reported during the summer smog condition (Saathoff et al., 2001),





but with lack of confirmation on a global scale. Moreover, the heterogeneous effects of $RO_2$ have been investigated only insufficiently (Jacob, 2000). In this study, we examine these uncertainties using the global model CHASER to perceive the respective and total effects of the HRs of $N_2O_5$, $HO_2$, and $RO_2$ on the tropospheric chemistry. For the interface of HRs in the

atmosphere, we tentatively consider surfaces of cloud particles and those of aerosols and discuss details of its effects in this study. In the following text, the research method, including model description and configuration, is described in section 2. In section 3.1, our model is verified with available observations including ground stations, ship/aircraft and satellite measurements, particularly addressing the roles of the HRs. The global effects of $N_2O_5$, $HO_2$, and $RO_2$ uptake are discussed in section 3.2 to elucidate cloud-particles and aerosol effects. Section 3.3 will discuss sensitivities of tropospheric chemistry to

the magnitudes of HRs. Section 4 presents a summary and concluding remarks.

## 2 Method

### 2.1 Global chemistry model

The global chemistry model used for this study is CHASER (MIROC-ESM) (Sudo et al., 2002, 2007; Watanabe et al., 2011), which considers detailed photochemistry in the troposphere and stratosphere. The chemistry component of the model, based

on CHASER-V4.0, calculates the concentrations of 92 chemical species and 262 chemical reactions (58 photolytic, 183 kinetic, and 21 heterogeneous reactions including reactions on PSCs); more details on CHASER can be found in an earlier report of the literature (Morgenstern et al., 2017). Its tropospheric chemistry considers the fundamental chemical cycle of $O_x$–$NO_x$–$HO_x$–$CH_4$–CO along with oxidation of non-methane volatile organic compounds (NMVOCs). Its stratospheric chemistry simulates chlorine and bromine-containing compounds, CFCs, HFCs, OCS, $NO_2$, and the formation of polar stratospheric

clouds (PSCs) and heterogeneous reactions on PSC surfaces. In the framework of MIROC-Chem, CHASER is coupled with the MIROC-AGCM atmospheric general circulation model (ver. 4; Watanabe et al., 2011). The meteorological fields simulated by MIROC-AGCM were nudged toward the six-hourly NCEP FNL data. For this study, the spatial resolution of the model was set as T42 (about 2.8° × 2.8° grid spacing) in horizontal and L36 (surface to approx. 50 km) in vertical. Anthropogenic emissions for $O_3$ and aerosol precursors like $NO_x$, CO, VOCs, and $SO_2$ are specified using the HTAP-II inventory for 2008

(http://edgar.jrc.ec.europa.eu/htap_v2/), with biomass burning emissions derived from the MAC reanalysis system.

In the model, the aerosol concentrations for BC/OC, sea-salt, and soil dust are handled by the SPRINTAR module, which is also based on the CCSR/NIES AGCM (Takemura et al., 2000). The bulk thermodynamics for aerosols are applied, including $SO_4^{2-}$ chemistry ($SO_2$ oxidation with OH, $O_3$/$H_2O_2$, cloud-pH dependent) $SO_4^{2-}$-$NO_3^-$-$NH_4^+$ and $SO_4^{2-}$-dust interaction.

### 2.2 Heterogeneous reactions in the chemistry–climate model (CHASER)

The CHASER-V4 model considers HRs in both the troposphere and stratosphere. In this work, we particularly examine HRs in the troposphere. In the current version of CHASER, tropospheric HRs are considered for $N_2O_5$, $HO_2$, and $RO_2$, using uptake coefficients for the distinct surfaces of aerosols (sulfate, sea-salt, dust, and organic carbons) and cloud particles (liquid/ice) as





listed in Table 2. Although some other views incorporate the catalysis of transition metal ions (TMI) Cu(I)/Cu(II) and Fe(II)/Fe(III) for the $HO_2$ conversion on aqueous aerosols (Li et al., 2018; Mao et al., 2013; Taketani et al., 2012), this

mechanism remains uncertainties (Jacob, 2000). The TMI mechanism might lead to either $H_2O_2$ (Jacob, 2000) or $H_2O$ product (Mao et al., 2018). However, this may not cause any significant difference since recycling $HO_2$ from $H_2O_2$ is ineffective (Li et al., 2018). For this study, the uptake of $HO_2$ is affirmed with $H_2O_2$ as the product (Loukhovitskaya et al., 2009; Taketani et al., 2009), generally used in many atmospheric models such that this is not counted as a terminal sink for $HO_2$ (Jacob, 2000; Lelieveld and Crutzen, 1990; Morita et al., 2004; Thornton et al., 2008). The $RO_2$ uptakes are assumed with inert products, as

suggested by Jacob (2000). The heterogeneous pseudo-first-order loss rate $\beta$ for the species $i$ is given using the theory of Schwartz (Dentener and Crutzen, 1993; Jacob, 2000; Schwartz, 1986), in which it is simply treated with the mass transfer limitations operating two conductances represented free molecular and continuum regimes for tropospheric clouds and aerosols, in addition to using reactive uptake coefficient ($\gamma$) instead of the mass accommodation coefficient as

$$\beta_i = \sum_j (\frac{4}{v_i \gamma_{ij}} + \frac{R_j}{D_{ij}})^{-1} . A_j \qquad (1)$$

Therein, $v_i$ stands for the mean molecular speed (cm s$^{-1}$) of species $i$, $D_{ij}$ is the gaseous mass transfer (diffusion) coefficient (cm$^2$ s$^{-1}$) of species $i$ for particle type $j$, and $A_j$ expresses the surface area density (cm$^2$ cm$^{-3}$) for particle type $j$. In the model, the particle size and effective radius $R_j$ for aerosols are calculated as a function of RH (Takemura et al., 2000). The aerosol concentrations are based on SPRINTAR for BC/OC, sea-salt, and dust (Takemura et al., 2000). The surface area density (SAD) for aerosols ($A_j$) is estimated using lognormal distributions of particle size ($SF_j$) with mode radii variable with the RH (Sudo

et al., 2002) as

$$A_{j,ae} = C_N * 4\pi R_j^2 * SF_j , \qquad (2)$$

where $C_N$ represents number density (cm$^{-3}$), $R_j$ signifies the effective radii (cm) of particle type $j$. To calculate SAD for cloud particles, the liquid water content (LWC) and ice water content (IWC) in the AGCM are converted using the cloud droplet distribution of Battan and Reitan (1957) and the relation between IWC and the surface area density for ice clouds (Lawrence

and Crutzen, 1998; McFarquhar and Heymsfield, 1996).

$$A_c = 10^{-4} * IWC^{0.9}$$

$$A_{j,ice} = 3 * A_c , \qquad (3)$$

In those equations, $A_c$ represents the cross-section area for ice crystals (cm$^2$ cm$^{-3}$). For liquid clouds, the following holds.

$$A_{j,liq} = LWC * 10^{-6} * \frac{3}{R_j} \qquad (4)$$

The uptake coefficient parameter ($\gamma$) is defined as the net probability that a molecule X undergoing a gas-kinetic collision with a surface is actually taken up onto the surface. Although several recent model studies that consider dependency of $\gamma$ on RH and/or T, majority of the earlier studies uses constant $\gamma$ values which only vary with aerosol particle compositions (Chen et al., 2018; Evans and Jacob, 2005; Macintyre and Evans, 2010, 2011). For one study, $\gamma_{HO_2}$ for the uptake onto aqueous aerosols is





considered with pH dependence (Thornton et al., 2008). However, another study demonstrated that the uptake is large,

irrespective of the solubility in cloud water or pH (Morita et al., 2004). Therefore, we instead choose $\gamma_{HO_2}$ as fix values depending on the type of particle. Indeed, from Eq. (1) it is apparent that uptake coefficients should be unimportant for uptake onto large particles such as cloud droplets. In this study, $\gamma$ for cloud particles of liquid and ice phases are given based on suggestions from earlier reports (Dentener and Crutzen, 1993; Jacob, 2000). One study (Dentener and Crutzen, 1993) used a constant $\gamma_{N_2O_5}$ of 0.1 for uptake on seasalt, sulfate, and cloud particles. They also revealed that smaller $\gamma_{N_2O_5}$ of 0.01, which

had been reported as laboratory measurements, has insensitivity to effects on tropospheric oxidant components. Results of another study (Jacob, 2000) indicated constants $\gamma_{N_2O_5} = 0.1$ and $\gamma_{HO_2} = 0.2$ for uptakes on both liquid clouds and aerosols, the later aims to involve the HO$_2$ scavenging by clouds without accounting for details of aqueous-phase chemistry. For ice crystals, Jacob suggested $\gamma_{HO_2} = 0.025$ based on a report by Cooper and Abbatt (1996). Jacob recommended using $\gamma_{RO_2} = 0.1$ for hydroxy-RO$_2$ group produced by oxidation of unsaturated hydrocarbons and $\gamma_{RO_2} = 4 \times 10^{-3}$ for PA. The $\gamma$ values for aerosols

are assumed to be fundamentally the same as those for liquid cloud particles in this study. It is noteworthy that the $\gamma$ values for cloud particles are given tentatively in this study and are adjusted based on evaluation of the resulting species concentrations of O$_3$, NO$_y$, and OH with the observations.

## 2.3 Experiment setup

In this study, simulations of two types were conducted to isolate the distinct effects of each HR for the surface types considered

in the model (Table 3 and Table S 1). Whereas a control simulation STD considers all HRs, noHR cases intentionally ignore one or all of the HRs to calculate effects of individual HRs. The sensitivity runs that turned off the separate HRs onto clouds (liquid and ice) and aerosols were also added to exploit the separate aerosol-heterogeneous and cloud-heterogeneous effects, as suggested in many earlier studies (Apodaca et al., 2008; Jacob, 2000; Lelieveld and Crutzen, 1990, 1991; Morita et al., 2004). The HR effects are determined as the differences between noHR cases and STD simulation as Eq. (5).

$$Impact(i)_j = \frac{(STD_i - noHR(j)_i)}{noHR(j)_i} * 100 \ (\%) \qquad (5)$$

Therein, STD$_i$ stands for the concentration of investigated atmospheric component $i$ in the STD run; and noHR$(j)_i$ denotes the concentration of component $i$ in the sensitivity run in which the HRs of/onto $j$ was ignored ($j$ could be N$_2$O$_5$, HO$_2$, RO$_2$, clouds, aerosols).

An additional sensitivity test was run to examine the sensitivity of the troposphere's responses with the amplified HRs

magnitudes (Table S 1). These simulations only apply for HR(N$_2$O$_5$) and HR(HO$_2$) to verify some uncertainties that have been argued among earlier studies (Chen et al., 2018; Evans and Jacob, 2005; Macintyre and Evans, 2010, 2011).





**Table 1: Computation packages in the chemistry-climate model "CHASER"**

| Base model | MIROC4.5 AGCM |
|---|---|
| Spatial resolution | Horizontal, T42 (2.8° × 2.8°); vertical, 36 layers (surfaces approx. 50 km) |
| Meteorology (u, v, T) | Nudged to the NCEP2 FNL reanalysis |
| Emission (anthropogenic, natural) | Industry traffic, Vegetation Ocean<br>Biomass burning specified by MAC reanalysis |
| Aerosol | BC/OC, sea-salt, and dust<br>BC aging with $SO_x$/SOA production |
| Chemical process | 94 chemical species, 263 chemical reaction (gas phase, liquid phase, non-uniform $O_x$-$NO_x$-$HO_x$-$CH_4$-CO chemistry with VOCs<br>$SO_2$, DMS oxidation (sulfate aerosol simulation)<br>$SO_4$-$NO_3$-$NH_4$ system and nitrate formation<br>Formation of SOA<br>BC aging<br>(+) Heterogeneous reactions: 8 reactions of $N_2O_5$, $HO_2$, $RO_2$ ; constant uptake coefficients ($\gamma$) on types of aerosols (Ice, Liquid, Sulfate, Sea salt, Dust, OC) |

**Table 2: Heterogeneous reactions in CHASER**

| No | Reactions | $\gamma_{ice}$ | $\gamma_{liq}$ | $\gamma_{sulf}$ | $\gamma_{salt}$ | $\gamma_{dust}$ | $\gamma_{oc}$ |
|---|---|---|---|---|---|---|---|
| 1 | $HO_2 \rightarrow 0.5H_2O_2 + 0.5O_2$ | 0.02 | 0.1 | 0.1 | 0.1 | 0.1 | 0.1 |
| 2 | $N_2O_5 \rightarrow 2HNO_3$ | 0.01 | 0.08 | 0.1 | 0.1 | 0.1 | 0.1 |
| $RO_2 \rightarrow$ products: | | | | | | | |
| 3 | $HOC_2H_4O_2 \rightarrow$ product* | 0.02 | 0.2 | 0.2 | 0.2 | 0.2 | 0.2 |
| 4 | $HOC_3H_6O_2 \rightarrow$ product* | 0.02 | 0.2 | 0.2 | 0.2 | 0.2 | 0.2 |
| 5 | $ISO_2 \rightarrow$ product* | 0.01 | 0.1 | 0.1 | 0.1 | 0.1 | 0.1 |
| 6 | $MACRO_2 \rightarrow$ product* | 0.01 | 0.1 | 0.1 | 0.1 | 0.1 | 0.1 |
| 7 | $CH_3COO_2 \rightarrow$ product* | 0 | 0.001 | 0.004 | 0.004 | 0.004 | 0.004 |

References and the mention of adjustments are given in the main text. The $RO_2$ uptakes are assumed with inert products as suggested by Jacob (2000).

**Table 3: Main sensitivity simulations for HRs in this work**

| No. | Simulation ID | HR: $N_2O_5$ | HR: $HO_2$ | HR: $RO_2$ | HRs on clouds | HRs on aerosols |
|---|---|---|---|---|---|---|
| 1 | STD | x | x | x | | |
| 2 | noHR | | | | | |
| 3 | noHR.n2o5 | | x | x | | |
| 4 | noHR.ho2 | x | | x | | |
| 5 | noHR.ro2 | x | x | | | |
| 6 | noHR.Cld | | | | | x |
| 7 | noHR.Ae | | | | x | |



### 2.4 Observation data for model evaluation

Model simulations with and without HRs are evaluated distinctively with stationary, ship-based, aircraft-based, and satellite-based measurements. The observational information and locations of the surface site and ship/aircraft tracks for the observations used for this study are summarized in Table 4 and Fig. 1.

EANET is well known as the Acid Deposition Monitoring Network in eastern Asia. The monthly data from 45 stations over 13 countries during 2010–2016 were used to verify surface concentrations of aerosols (sulfate, nitrate) and trace gases

($HNO_3$, $NO_x$, $O_3$) in eastern Asia. We also used data of the European Monitoring and Evaluation Programme (EMEP), which compile observations over 245 European stations.

Additionally, we exploited ship-based observational data from R/V MIRAI cruise (http://www.godac. jamstec.go.jp/darwin/e) undertaken by the Japan Agency for Marine-Earth Science and Technology (JAMSTEC). This study used data for surface CO and $O_3$ concentrations in summer 2015–2017 along the Japan–Alaska, Japan–Indonesia–Australia

routes (Kanaya et al., 2019). The model data were compiled in hourly time-steps and were interpolated corresponding with the MIRAI time step and coordinates. For verification of the vertical tropospheric profiles, we used Atmospheric Tomography (ATom1) aircraft measurements (https://espo.nasa.gov/atom/content/ATom) for $NO_2$, OH, CO, and $O_3$.

The simulated tropospheric ozone was also evaluated using the tropospheric column $O_3$ (TCO) derived from the OMI satellite data (https://daac.gsfc.nasa.gov/). For distribution of the cloud fraction, satellite data from International Satellite Cloud

Climatology Project (ISCCP, https://isccp.giss.nasa.gov/), GCM-Oriented CALIPSO Cloud Products (CALIPSO-GOCCP, https://eosweb.larc.nasa.gov/project/calipso/calipso_table), and Japanese 55-year reanalysis (JRA-55 - https://doi.org/ 10.5065/D6HH6H41) were used.

Model bias and normalized root mean squared error (NRMSE) for each species were calculated as shown below, where $n$ is the number of available data (number of stations × time-step).

$$\text{bias} = \frac{\sum_1^n \text{Model} - \text{Observation}}{n} \tag{6}$$

$$\text{NRMSE} = \frac{\sqrt{\frac{\sum_1^n (\text{Model} - \text{Observation})^2}{n}}}{\text{Observation}} \tag{7}$$





**Figure 1: Locations of EANET stations (a), EMEP stations (b), MIRAI cruises (c), and ATom1 flights (d).**
**Source: (a) https://monitoring.eanet.asia/document/overview.pdf; (b) https://projects.nilu.no/CCC/**

**Table 4: Datasets used for verification in this study**

| Verified species | Regions | Data | Time series | Time-step |
|---|---|---|---|---|
| Sulfate, nitrate, $NO_x$, $O_3$, $HNO_3$ | Eastern Asia | EANET | 2010–2016 | Daily to 2-weekly |
| Sulfate, nitrate, $NO_x$, $O_3$, CO | Europe | EMEP | 2010–2016 | Hourly |
| CO, $O_3$ | Surface of the Pacific Ocean (Australia – Indonesia – Japan – Alaska) | MIRAI | 8,9/2015 1,8,9/2016 7,8,9/2017 | 30 min |
| $NO_2$, OH, CO, $O_3$ | Various altitudes above the Pacific and Atlantic Ocean | ATOM1 | 8/2016 | 30 min |
| TCO | 60S–60N (Satellite) | OMI | 2010–2016 | Daily |
| Cloud fraction | Global (Satellite) | ISCCP | 2000–2009 | Monthly |
| | Global (Satellite) | CALIPSO-GOCCP | 2007–2017 | Monthly |
| | Global (Reanalysis) | JRA55 | 2000–2015 | 6-hourly |





## 3 Results and Discussion

### 3.1 Model verifications

### Cloud verification

For this study, we tentatively consider HRs on the cloud particle surface. Given the great uncertainties related to the reaction coefficient (γ) (Macintyre and Evans, 2010, 2011), the cloud distributions must be examined adequately in the model to the greatest extent possible. The model-calculated cloud distributions were verified using satellite observation data ISCCP D2, CALIPSO-GOCCP, and reanalysis data JRA55.

For the entire troposphere, the calculated cloud fraction was generally underestimated against the satellite observations 230 and reanalysis data (Fig. 2, the first row). At the North Pacific region in JJA (Fig. 2, the second row), when the cloud fraction peaked in the region, the model was able to reproduce the satellite observations (ISCCP and CALIPSO). However, for the lower troposphere over the region, the cloud fraction calculated using CHASER in JJA appears to be overestimated (Fig. 2, the fourth row), suggesting that the resulting HR effects would also be exaggerated to some extent.

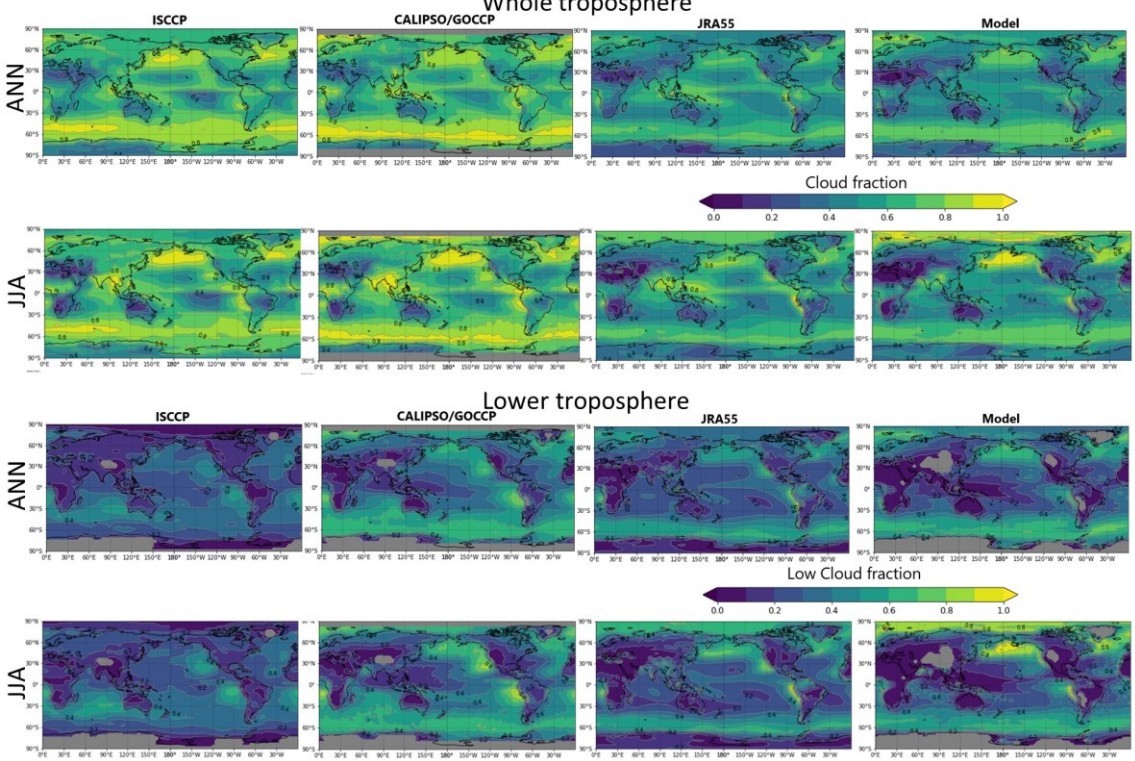

**Figure 2: Comparisons for cloud fraction in the whole troposphere (first and second rows) and lower troposphere (third and fourth rows). ANN denotes annual mean; JJA denotes June + July + August mean. First column for ISCCP (2000–2009), second column for CALIPSO/GOCCP (2007–2017), and third and fourth columns for JRA55 and CHASER (2000–2015). Color-bars are the same for all panels. In ISCCP and CALIPSO data, the pressure boundary layer of the low troposphere is > 680 hPa. In JRA55, the low troposphere was defined as 850–1100 hPa of pressure.**





**Verification with stationary observations**

Verifications with EANET and EMEP stationary observations were conducted to assess the model performance on land domains of eastern Asia and Europe, particularly addressing the roles of the heterogeneous reactions considered for this study. The mass concentrations of particulate matter ($PM_{2.5}$), sulfate ($SO_4^{2-}$), nitrate ($NO_3^-$), aerosols and gaseous $HNO_3$, $NO_x$, $O_3$, and CO (CO only for EMEP) of 2010–2016 were evaluated (see Fig. S3 to S10 for monthly concentrations and Fig. S12: for

correlations). In general, the model can moderately reproduce the $PM_{2.5}$, $SO_4^{2-}$, and $NO_3^-$ aerosol concentrations at these locations ($R$ = 0.3–0.7, Table 5), although $PM_{2.5}$ was underestimated, sulfate was overestimated slightly. Nitrate was underestimated for EANET and overestimated for EMEP. It is noteworthy that the model performance for EMEP stations was better than that for EANET. The $PM_{2.5}$ concentration was better estimated with the inclusion of $N_2O_5$ and $HO_2$ uptakes (bias reduction in Table 5). The high negative biases for $NO_3^-$ are significant at urban sites, e.g. at Tokyo (Fig. 3), which can be

associated with undervaluation for $NO_x$ and which can thereby lessen the effects of $N_2O_5$ uptake.

Nitric acid in both regions was overestimated. The correlations, biases, and normalized root mean square error (NRMSE) of the model for $SO_4$, $NO_3$, and $HNO_3$ are in the ranges as reported in a multi-model study by Bian et al. (2017) (Table 6).

The $NO_x$ concentration for eastern Asia and Europe was underestimated, with significant bias for Asian polluted locations. The increasing effects of $NO_x$ attributable to heterogeneous reactions, although minor, mitigated these underestimations. In

Fig. 3, although $NO_x$ was partly reduced via uptake of $N_2O_5$, the $NO_x$ level was mostly increased because of $HO_2$ and $RO_2$ uptake. CO for EMEP was underestimated by the model. This undervalue was mitigated by increasing effects because of HRs of $N_2O_5$ and $HO_2$. The uptakes of $RO_2$, in contrast, minorly reduced CO levels so that the model bias was worsened slightly. For $O_3$, whereas the model tends to overestimate this tracer for both regions, $O_3$ reduction effects of all HRs also alleviated the model overestimates, especially in June, July, and August (JJA). In December, January, and February (DJF), the model tended

to underestimate $O_3$ levels at some stations. The reduction effects on $O_3$ extended this undervalue. In general, STD simulation with coupled HRs partly improved the agreement related to the particulate and gaseous species, showing less bias than that of simulations without HRs (Table 5).





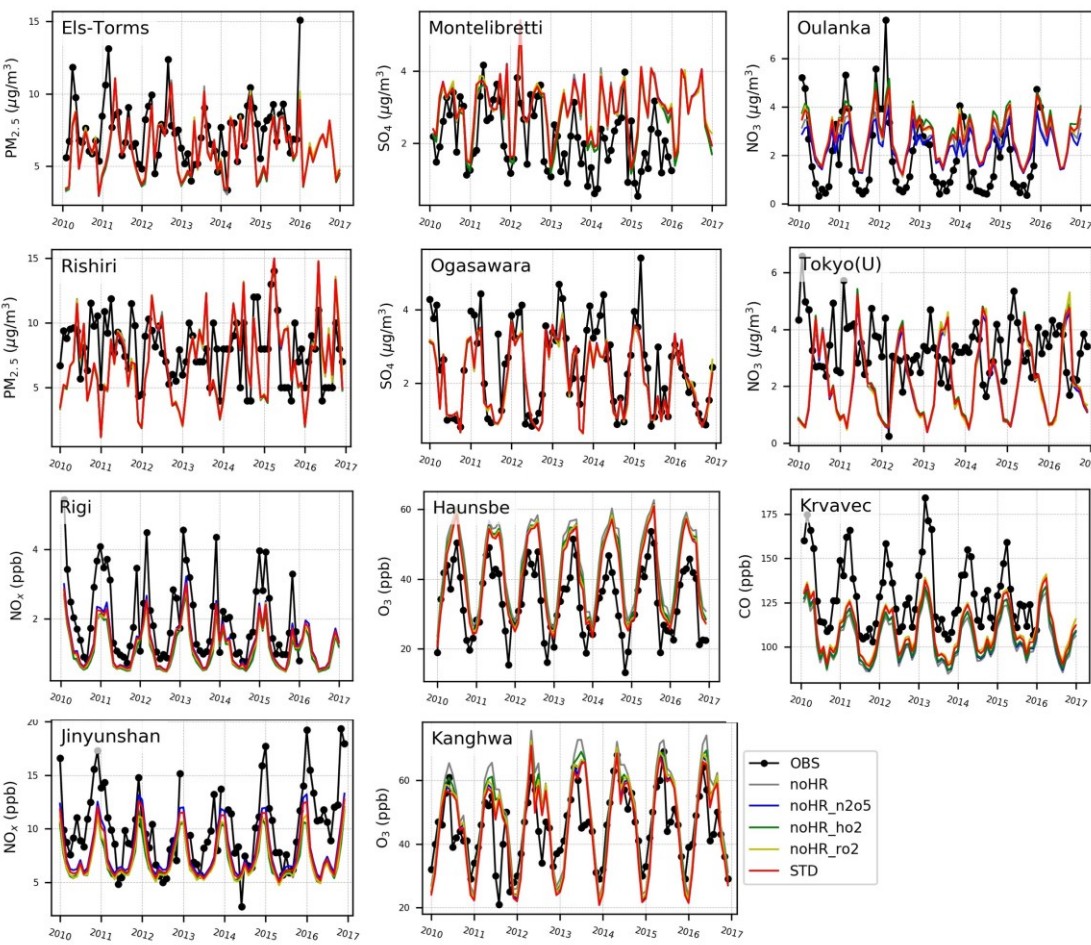

**Figure 3: Observations and sensitivity simulations at represented EMEP and EANET stations: first row – PM₂.₅, SO₄²⁻, and NO₃⁻ for EMEP; second row – same as first row but for EANET; third row – NOₓ, O₃, CO for EMEP; fourth row – NOₓ and O₃ for EANET.**

**Table 5: Model correlations and biases with EANET/EMEP observations: three-sigma-rule outlier detection is applied for each station before calculating all data. For NOₓ, all data were filtered once more using the two-sigma-rule. Bias of the sensitivity run is shown as bold if it is higher than the bias of the STD run.**

| | EANET | | | | | | EMEP | | | | | | |
|---|---|---|---|---|---|---|---|---|---|---|---|---|---|
| | PM$_{2.5}$ | SO$_4^{2-}$ | NO$_3^-$ | HNO$_3$ | NO$_x$ | O$_3$ | PM$_{2.5}$ | SO$_4^{2-}$ | NO$_3^-$ | HNO$_3$ | NO$_x$ | O$_3$ | CO |
| R(STD) | 0.37 | 0.56 | 0.379 | 0.177 | 0.233 | 0.6 | 0.475 | 0.633 | 0.715 | 0.116 | 0.698 | 0.651 | 0.534 |
| bias (STD) | -7.526 | 1.048 | -0.395 | 0.311 | -3.929 | 3.927 | -2.966 | 0.784 | 0.273 | 0.081 | -0.773 | 4.071 | -3.439 |
| bias (noHR) | -7.442 | 0.971 | **-0.452** | 0.292 | **-4.011** | 6.808 | **-3.262** | 0.603 | 0.106 | 0.067 | **-0.895** | 7.189 | **-9.062** |
| bias (noHR_n2o5) | **-7.575** | **1.05** | **-0.46** | 0.295 | -3.869 | **4.93** | **-3.223** | 0.774 | 0.042 | 0.07 | -0.707 | **5.013** | **-6.822** |
| bias (noHR_ho2) | **-7.607** | 0.925 | -0.37 | **0.312** | **-4.02** | 5.126 | **-3.136** | 0.55 | **0.335** | 0.078 | **-0.895** | 5.489 | **-6.896** |
| bias (noHRs_ro2) | -7.38 | 1.021 | **-0.427** | 0.305 | **-4.008** | 4.931 | -2.858 | **0.839** | 0.275 | 0.079 | **-0.833** | 4.893 | -2.276 |





**Table 6: Comparisons between EANET and EMEP observations with atmospheric models. Outlier detection follows in Table 5. The model result is shown as bold if it is better than or agreed with Bian's report.**

| EANET | $SO_4$ [$\mu g\ m^{-3}$] | $NO_3$ [$\mu g\ m^{-3}$] | $HNO_3$ [ppb] |
|---|---|---|---|
| This study | $r$ = **0.56** | $r$ = **0.379** | $r$ = **0.177** |
| | bias = **1.048** | bias = **-0.395** | bias = **0.311** |
| | nrmse = **0.954** | nrmse = **1.58** | nrmse = **2.491** |
| Bian et al., 2017 | $r$ = 0.449–0.640 | $r$ = 0.226–0.448 | $r$ = 0.098–0.370 |
| | bias = 0.358–1.353 | bias = 0.338–1.920 | bias = 0.347–3.596 |
| | nrmse = 0.840–0.968 | nrmse = 1.494–2.080 | nrmse = 0.980–2.880 |
| EMEP | $SO_4$ [$\mu g\ m^{-3}$] | $NO_3$ [$\mu g\ m^{-3}$] | $HNO_3$ [$\mu g\ m^{-3}$] |
| This study | $r$ = **0.633** | $r$ = **0.715** | $r$ = 0.116 |
| | bias = **0.784** | bias = **0.273** | bias = **0.886** |
| | nrmse = 0.961 | nrmse = **0.91** | nrmse = 3.33 |
| Bian et al., 2017 | $r$ = 0.230–0.463 | $r$ = 0.393–0.585 | $r$ = 0.157–0.502 |
| | bias = 0.452–1.699 | bias = 0.539–1.421 | bias = 0.570–3.836 |
| | nrmse = 0.514–0.818 | nrmse = 0.745–1.031 | nrmse = 0.908–2.542 |

**Verification with ship-based measurements**

The model simulations were also verified with $O_3$ and CO observations from the Research Vessel (R/V) MIRAI for the Pacific Ocean region. This study specifically examines data from the four cruises of R/V MIRAI for the Japan–Alaska region (40° N–75° N, 140° E–150° W) in summer, designated as Track 1 (2015/8/28–9/29), Track 4 (2016/8/22–9/14), Track 5 (2017/7/11–7/29), and Track 6 (2017/8/24–9/8), and cruises during DJF for the Indonesia–Australia region (5°–25° S, 105–115° E) designated as Track 2 (2015/12/23–2016/1/10) and for the Indonesia–Japan region (10–35° N, 129°–140° E) designated as

Track 3 (2016/1/17–1/24). Data for the North Pacific region (40°–60° N) are addressed for analysis in Sect. 3.2.

Table 7 shows correlation coefficients (plotted in Fig. S13), indicating that the CHASER simulations for CO and $O_3$ are in good agreement with MIRAI observations ($R$ = approx. 0.6). However, the model still shows some discrepancies for both CO and $O_3$ concentrations. In general, the estimated CO and $O_3$ are both reduced for T1, T4-6 as compared to observations whereas superior for the data located in 20° S–20° N during T2-3. Overestimations for CO and $O_3$ occurring in the region with

considerably low levels of these species might be attributed to the lack of halogen chemistry in the model, as also discussed for the nearby region in a past report (Kanaya et al., 2019). Undervalues for CO and $O_3$ levels in the higher latitudes (T1, T4–6) are ascribable to the insufficient downward mixing process of stratosphere $O_3$ in the model (Kanaya et al., 2019).

The negative biases in the noHR simulations for CO are lower in the STD run for all cruises, as they are for the North Pacific region (second versus third/fourth/fifth data rows for CO, Table 7). The CO-increasing effects by $N_2O_5$ and $HO_2$ uptakes

are consistent with the comparison for EMEP. So are CO-reduction effects because of HRs($RO_2$). Whereas the effects by $N_2O_5$ and $HO_2$ reduce the model bias, the CO-reducing effects by HRs($RO_2$) exaggerated the CO bias (second versus sixth data rows for CO in Table 7), which is already apparent for comparison with EMEP (last column, Table 5).

For $O_3$ level, the model undervalues (Table 7) are in the opposite direction to the $O_3$ overestimates for EANET and EMEP stations (Table 5). The lower panels presented in Fig. 4 show marked $O_3$ reduction with all HRs, mostly contributed from the

$HO_2$ uptake onto cloud particles (gaps between red and green lines). This marked reduction of the $O_3$ level is evident at some



points during the cruises, especially in the North Pacific region (the shaded areas), especially for T4. Unlike comparisons for land-domain data (Table 5), $O_3$ reduction because of HRs extends the model underestimates during the MIRAI cruises. It is noteworthy that one cannot necessarily confirm whether the STD run better simulates these species than the noHR does because tropospheric CO and $O_3$ levels are controlled by a complicated chemical mechanism and an interplay of emission, transport,

deposition, and local mixing in the boundary layers. As discussed later in Sect. 3.2, the surface aerosols concentration in the West Pacific Ocean mostly dominated by liquid clouds (exceed 50,000 $\mu m^2$ $cm^{-3}$ during JJA) and sulfate aerosols (approximately 75 $\mu m^2$ $cm^{-3}$ in JJA). The model improvements in reproducing CO by adding $N_2O_5$ and $HO_2$ uptake indicate that the appropriate mechanisms of these processes onto cloud droplets and sulfate aerosols are well-established in the model. For HRs($RO_2$), which induce the smallest and opposite effects on CO compared with the effects of $N_2O_5$ and $HO_2$ uptakes, it

can be stated in general for the total HR effects that including all three HRs partially improves the model during MIRAI cruises.

**Table 7: Model correlations and biases for MIRAI. No outlier filtration is applied. The bias of the sensitivity run is shown as bold if it is higher than the bias of STD run. The unit for the biases is ppb.**

|  | CO | $O_3$ | CO (40–60° N) | $O_3$ (40–60° N) |
|---|---|---|---|---|
| R(STD) | 0.689 | 0.617 | 0.58 | 0.665 |
| bias (STD) | -4.988 | -4.996 | -11.668 | -3.493 |
| bias (noHR) | **-10.324** | -2.388 | **-17.625** | 1.211 |
| bias (noHR_n2o5) | **-8.127** | -4.362 | **-14.804** | -2.738 |
| bias (noHR_ho2) | **-9.036** | -3.358 | **-16.358** | -0.226 |
| bias (noHR_ro2) | -3.433 | -4.431 | -10.035 | -2.526 |
| bias (noHR_cld) | **-6.821** | -3.199 | **-14.005** | 0.025 |

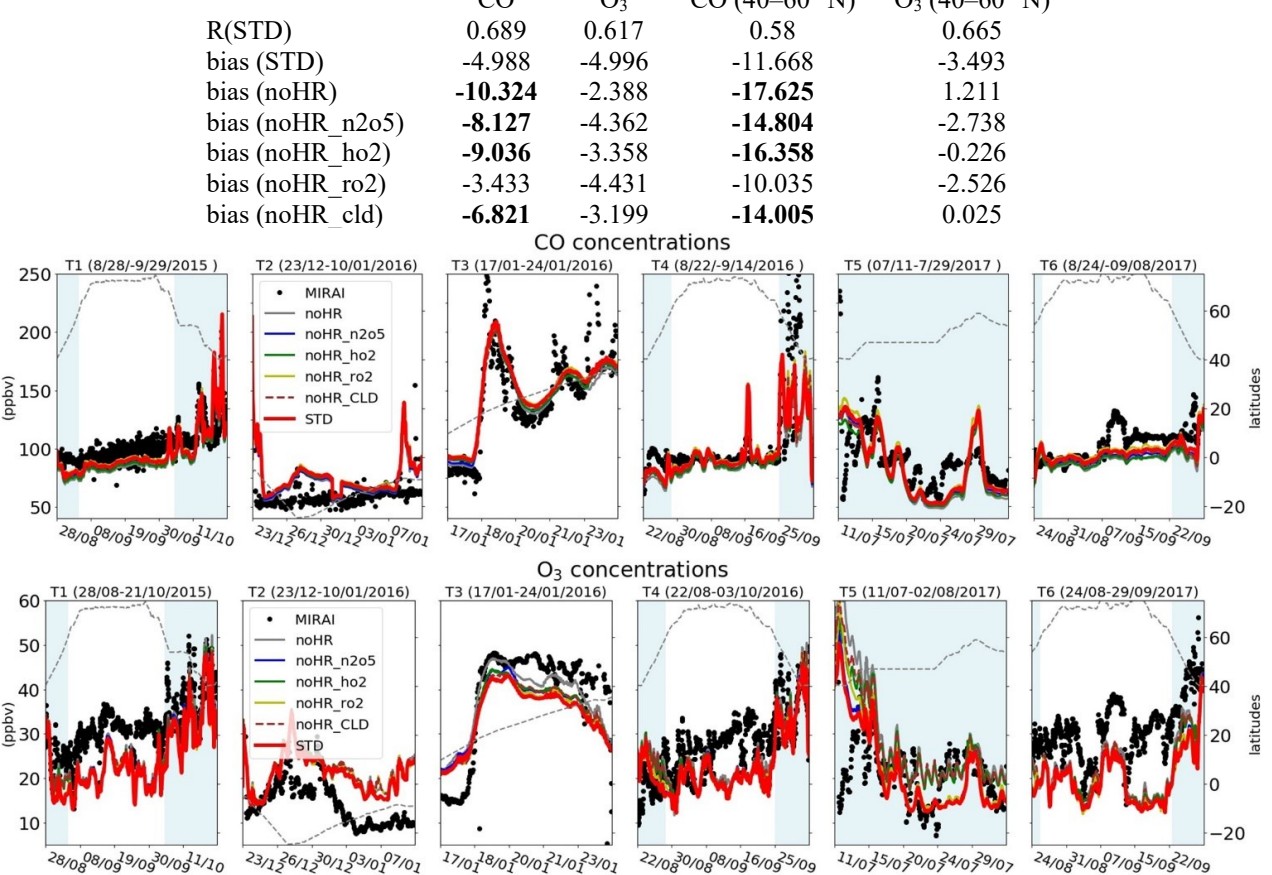

**Figure 4: Sensitivity simulations for CO (upper panels) and $O_3$ (lower panels) during MIRAI cruises. The left axis shows**
**concentrations. Dashed lines show latitudes of cruises scaled with the right axis. The horizontal axis shows cruise travel times. Shaded areas show data in the North Pacific region (140–240° E, 40–60° N). Brown dashed lines representing the noHR_CLD run are discussed in Sect. 3.2.**





**Verification using aircraft measurements**

To verify the model performance in the free troposphere, we used ATom1 aircraft measurements in August of 2016 (for NO₂,
OH, CO, and O₃). The spatial and temporal concentrations are available in Fig. S11. Correlations are shown in Fig. S14.

In general, the model simulations for NO₂, OH, CO, and O₃ adequately agree with aircraft measurements with $R$>0.5 (Fig.
S14). However, NO₂ and CO still tend to be underestimated by the model, which is consistent with comparisons for
EANET/EMEP and MIRAI observations. In Fig. 5, the CO-increasing effects, mostly by the uptake of $N_2O_5$ and $HO_2$,
mitigated the negative bias of the model. This CO bias reduction was visible for all flight altitudes, the lower troposphere, and
North Pacific region (Table 8). Both $N_2O_5$ and $HO_2$ uptakes show improvements for CO reproduction of the model. However,
$RO_2$ uptake seems to worsen the model's CO bias, which is consistent with the MIRAI comparison.

For the $O_3$ level, the model generally overestimates $O_3$ when calculating for all altitudes or lower troposphere, which is
similar to the EANET/EMEP observations. In the North Pacific region with $P > 600$ hPa (40–60° N, 198–210° W), the model
bias for $O_3$ in STD run turns to underestimate (second data row – second column from the right, Table 8), which might be
similar with MIRAI data verification for the western North Pacific (143° E–193° W). However, for the underlayers (>700 hPa)
show overestimation again (second data row – last column, Table 8). As MIRAI and Atom1 data show, the underestimates for
$O_3$ exist at the marine boundary layer in the western North Pacific and extend to the upper troposphere (<700 hPa) of the east
side, might be ascribed to the insufficient downward mixing process of stratosphere $O_3$ in the model as discussed previously.

The HR effects on $O_3$ are generally negative effects (all-flight mean concentration is 78.17 ppb by STD and 80.178 ppb
by noHR runs), although  they are small and barely recognizable in Fig. 5, which mitigates the model bias in the noHR run.
This model improvement is consistent for all flight altitudes, the low troposphere, and the North Pacific region (second versus
third data rows in Table 8). Both HR($N_2O_5$) and HR($RO_2$) typically contribute to this improvement (second versus fourth, fifth
data rows in Table 8). In contrast, HR($HO_2$) seems only to reduce the model bias in the ground layer, which is > 800 hPa for
all flights and > 700 hPa for the North Pacific region (second versus sixth rows in Table 8). At the bottommost layers in this
region, the model's overestimates for $O_3$ are reduced by the negative effects of $HO_2$ uptake. The extension of model bias
because of $HO_2$ uptake above 800 hPa is attributable to its increasing effect on $O_3$ level: the all-flight mean concentration is
78.17 ppb by STD and 77.96 ppb by noHR_ho2 runs. We recognize that this $O_3$ increase effect is opposite to the effects for
EANET/EMEP and MIRAI comparisons, which is discussed in Sect. 3.2 for $HO_2$ uptake effects.

The vertical means of model biases for all four species (NO₂, OH, CO, O₃) are presented in Fig. 6. In general, the STD
run reduces model biases for all four species, with better performance for broader regions (all flight-pressures and Northern
Hemisphere) than for the smaller region (North Pacific). In the North Pacific region, the negative bias for $O_3$ is observed only
for the 500–900 hPa layers (right-bottom panel of Fig. 6). The model bias is apparently extended in this region. However, the
inclusion of HR($HO_2$) reduces $O_3$ bias in this region (red line versus green line in the same panel), which might indicate that
the $O_3$ increase effect by HR($HO_2$) is verified particularly in 500–900 hPa layers during ATom1.

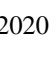



We also verify the total uptake of $N_2O_5$, $HO_2$, and $RO_2$ onto ice and liquid clouds using data obtained from ATom1 flights within the free troposphere. As Table 8 and Fig. 6 show, the inclusion of HRs onto clouds reduces the model biases for CO and $O_3$ in all calculations. The vertical-mean biases for $NO_2$, OH, CO, and $O_3$ species are all reduced by the inclusion of HRs onto clouds. However, Fig. 6 shows model worsening at 500–900 hPa, which coincides with the area in which $O_3$ is underestimated as described above. This result might prove that cloud overestimation for the North Pacific, as revealed at the

beginning of this section, affects the model bias in this region.

**Table 8: Model correlations and biases with ATom1: three-sigma-rule was applied for CO and $O_3$. NP denotes North Pacific region (140–240° E, 40–60° N). The bias of sensitivity run, which is higher than the bias of STD run, is presented as bold.**

|  | CO | $O_3$ | CO (>600 hPa) | $O_3$ (>600 hPa) | $O_3$ (>800 hPa) | CO (NP, >600 hPa) | $O_3$ (NP, >600 hPa) | $O_3$ (NP, >700 hPa) |
|---|---|---|---|---|---|---|---|---|
| R(STD) | 0.642 | 0.742 | 0.805 | 0.679 | 0.659 | 0.918 | 0.755 | 0.844 |
| Bias (STD) | -4.462 | 15.337 | -9.42 | 2.162 | 1.257 | -16.548 | -0.239 | 2.022 |
| Bias (noHR) | **-8.581** | **17.345** | **-13.589** | **3.266** | **2.365** | **-21.025** | **0.902** | **2.884** |
| Bias (noHR_n2o5) | **-7.583** | **16.697** | **-12.477** | **2.925** | **1.829** | **-19.44** | **0.32** | **2.44** |
| Bias (noHR_ho2) | **-6.101** | 15.127 | **-11.278** | 2.095 | **1.312** | **-18.247** | 0.035 | **2.526** |
| Bias (noHR_ro2) | -3.359 | **16.412** | -8.241 | **2.55** | **1.537** | -15.741 | **0.574** | **2.163** |
| Bias (noHR_cld) | **-4.978** | **16.141** | **-10.023** | **2.403** | **1.596** | **-17.199** | **0.725** | **2.904** |

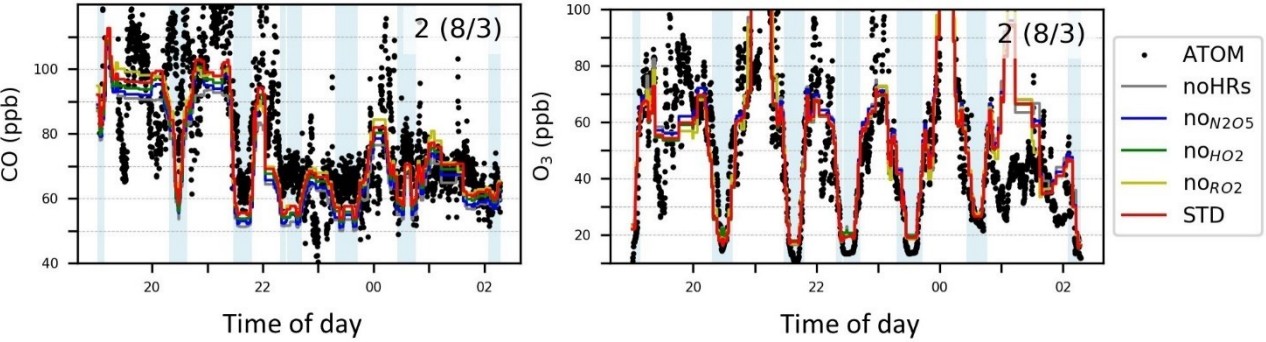

**Figure 5: Sensitivity simulations for CO and $O_3$ during Atom1 flight 2 (198–210° E, 20–62° N). Blue shaded areas show data for $P >$ 600 hPa.**



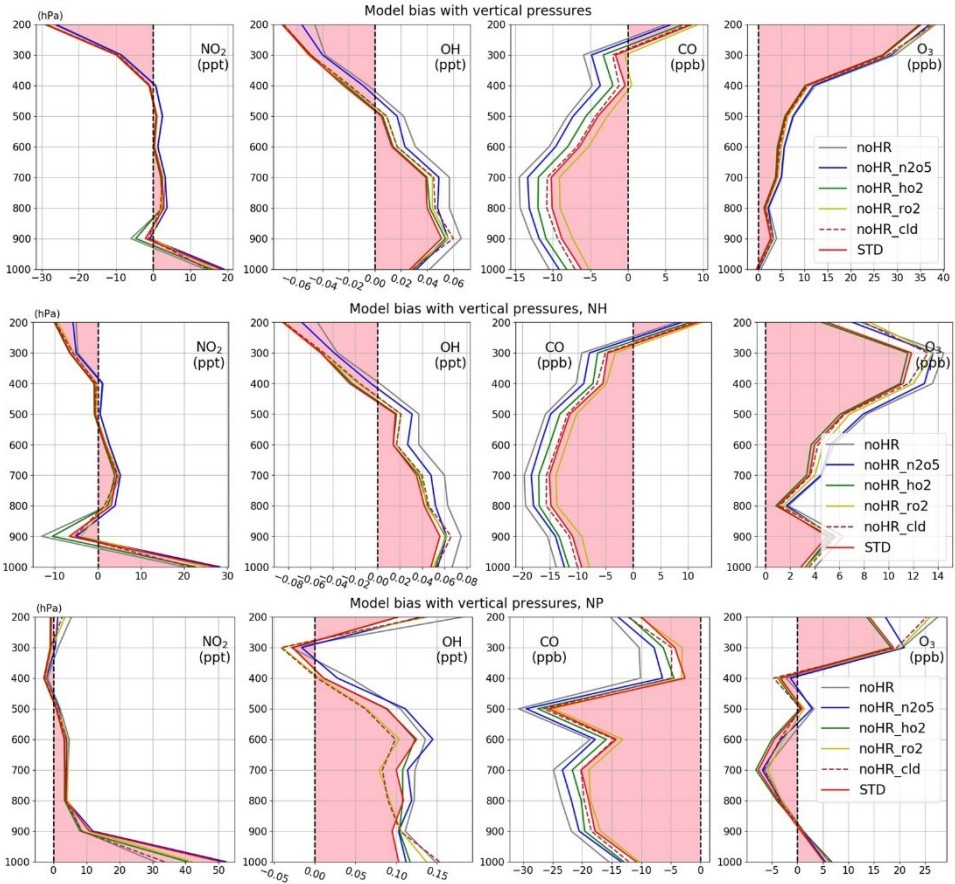

**Figure 6: Vertical bias against ATOM1. Data for each pressure level *P* are calculated within the range of *P*±50 hPa, with the applied three-sigma-rule for outlier detection. First, second, and third rows respectively show calculations for all flight domains, Northern Hemisphere, and North Pacific region. The horizontal axis shows model bias with units written in each panel. The vertical axis shows pressure (hPa). Brown dashed lines representing the noHR_cld run.**

## Verification with OMI satellite observation for TCO

We also tested STD and noHR simulations using the tropospheric column ozone (TCO) derived from the OMI satellite instrument (Fig. S1 and Fig. 7). In a large area of the Northern Hemisphere, the inclusion of HRs (STD run) generally improved the consistency with the OMI TCO (Fig. 7), particularly enhancing the winter minima (first and second panels in Fig. S1). This improvement in DJF is attributed mostly to the reductive effects of HR($N_2O_5$) and HR($RO_2$) in the lower (800 hPa) and middle troposphere (500 hPa), respectively (see Fig. 10 for vertical profiles of HR($N_2O_5$) on $O_3$ and Fig. 14 for vertical profiles of HR($RO_2$)). In the North Pacific, HRs appeared to extend $O_3$ underestimates, especially for latitudes higher than 40° N (Fig. 7) during the first half of the year (third panel, Fig. S1). However, such a discrepancy, which was also observed from comparison for R/V MIRAI observations (Fig. 4), might result from other factors such as deposition or vertical mixing rather than by HRs.



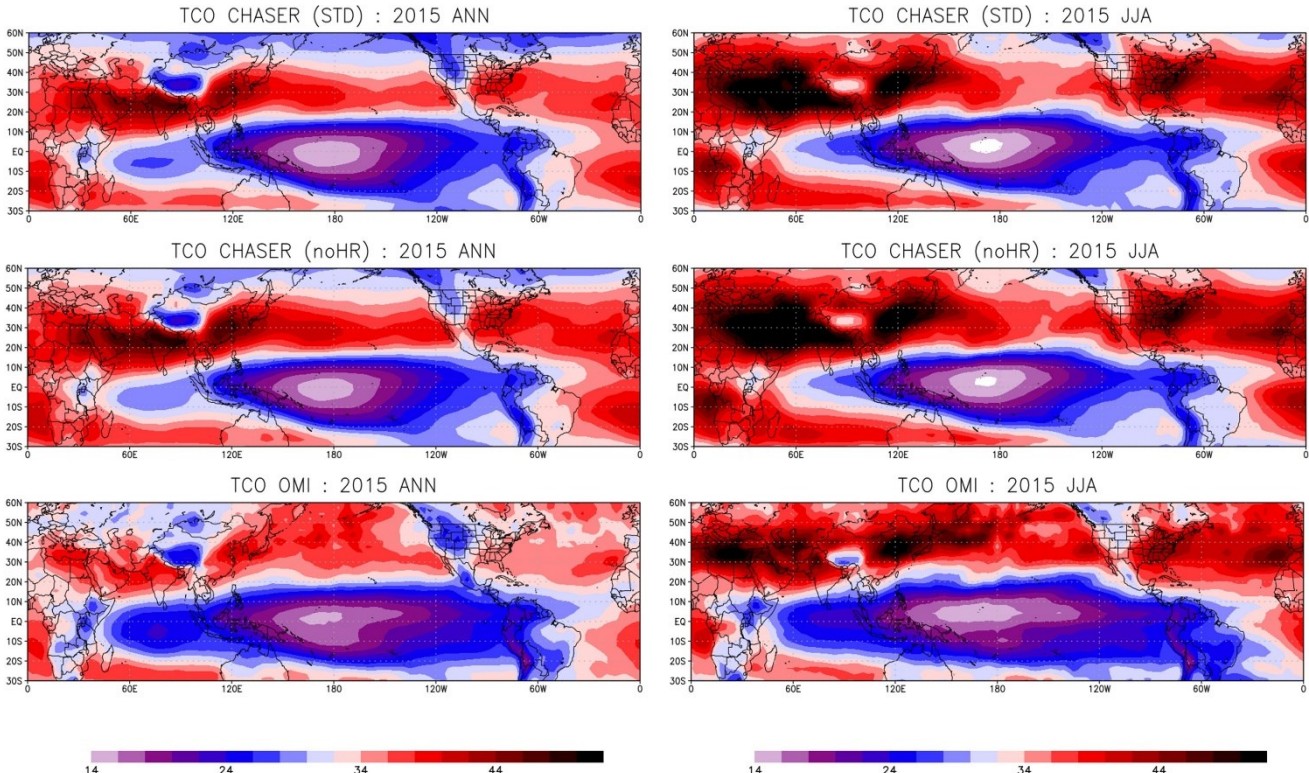

**Figure 7: Tropospheric column ozone (TCO) by CHASER (first and second rows) and OMI (third row): ANN is annual; JJA is June, July, and August.**

## 3.2 HR effects

This section presents a discussion of the global effects of HRs calculated using CHASER with their spatial distributions in the troposphere using standard (STD) and sensitivity simulations (noHR_n2o5, noHR_ho2, noHRs_ro2, noHR) for the meteorological year of 2011. Aside from the main simulations described in Table 3, additional runs that separately turned off the uptakes onto clouds or aerosols for each HR are also conducted to exploit the contributions of effects to the troposphere.

**Distribution of clouds and aerosols surface aerosol density (SAD)**

To obtain the parameters for uptake to clouds and aerosols, SAD estimations are used together with cloud fraction and aerosols concentration. Hereinafter, we discuss SAD distributions for total aerosol, ice clouds, and cloud droplets, which are estimated for the model using Eqs. (2), (3), and (4), respectively.

In Fig. 8, total surface area concentrations of liquid clouds and aerosols are both much lower aloft than at the surface (as counted on the dry and wet depositions). The liquid cloud SAD results are two orders of magnitude larger than ice cloud SAD and total aerosol SAD. The ice cloud SAD, distributed at the middle and upper troposphere, is enhanced for N/S middle latitudes in wintertime. Liquid cloud SAD concentrates mainly at the surface with distributions extending to 500 hPa, and





maximized at approx. 800 hPa over the mid-latitude storm tracks and in tropical convective systems, especially at 60° N in JJA. Total aerosol SAD was derived mainly from pollution sources at 40° N during both seasons with higher concentrations

apparent for DJF, with a greater spatial spread observed for JJA. Sulfate aerosols are becoming the dominant source of aerosol surface area in the model above 600 hPa (approx. 20 $\mu m^2$ $cm^{-3}$) in addition with organic carbons and soil dust (both are approx. 10 $\mu m^2$ $cm^{-3}$ in JJA) for the Northern Hemisphere.

In Fig. S2, showing the SAD distribution at the surface, the SAD for liquid clouds is dominant in JJA reaching approx. 50,000 $\mu m^2$ $cm^{-3}$ for middle-latitude and high-latitude ocean regions. Liquid clouds are the most contribution to the SAD at

the surface. Our model performance for aerosols SAD shows agreement with that presented in an earlier report (Thornton et al., 2008). Sulfate aerosols are prevalent in the northern mid-latitudes near industrial bases, maximize at the surface in DJF for the Chinese region (exceeding 1,000 $\mu m^2$ $cm^{-3}$), NE U.S. (approx. 500 $\mu m^2$ $cm^{-3}$), western Europe, and transport to the North Pacific region in JJA (approx. 250 $\mu m^2$ $cm^{-3}$). Soil dust aerosol SAD dominate in the regions of the Sahara and Gobi deserts, reaching annual average values exceeding 100 $\mu m^2$ $cm^{-3}$. Organic carbon (OC) is a dominant source of aerosol SAD over

biomass burning regions in China (up to 1,000 $\mu m^2$ $cm^{-3}$ in DJF), South Africa (up to 800 $\mu m^2$ $cm^{-3}$ in JJA), West Europe, and South America. The black carbon (BC) surface area can reach values exceeding 600 $\mu m^2$ $cm^{-3}$ in DJF for the region of China or other significant industrial areas (India reaches 75 $\mu m^2$ $cm^{-3}$, NE U.S., Europe) or over tropical forests, primarily in Africa. Sea salt aerosols are most important in high-latitude oceans during winter. However, the maximum contributions only reach 2 $\mu m^2$ $cm^{-3}$ in our model, which is much underestimated compared to Thornton's work (75 $\mu m^2$ $cm^{-3}$) (Thornton et al., 2008). In

brief, SAD for aerosols of all types contributes the most during DJF, whereas during JJA, the SAD for liquid clouds and sulfate aerosols are dominant, particularly for the northern high-latitude and mid-latitude oceans. The total aerosols SAD in this region are approx. 75 $\mu m^2$ $cm^{-3}$, which is consistent with estimation by Thornton (2008).

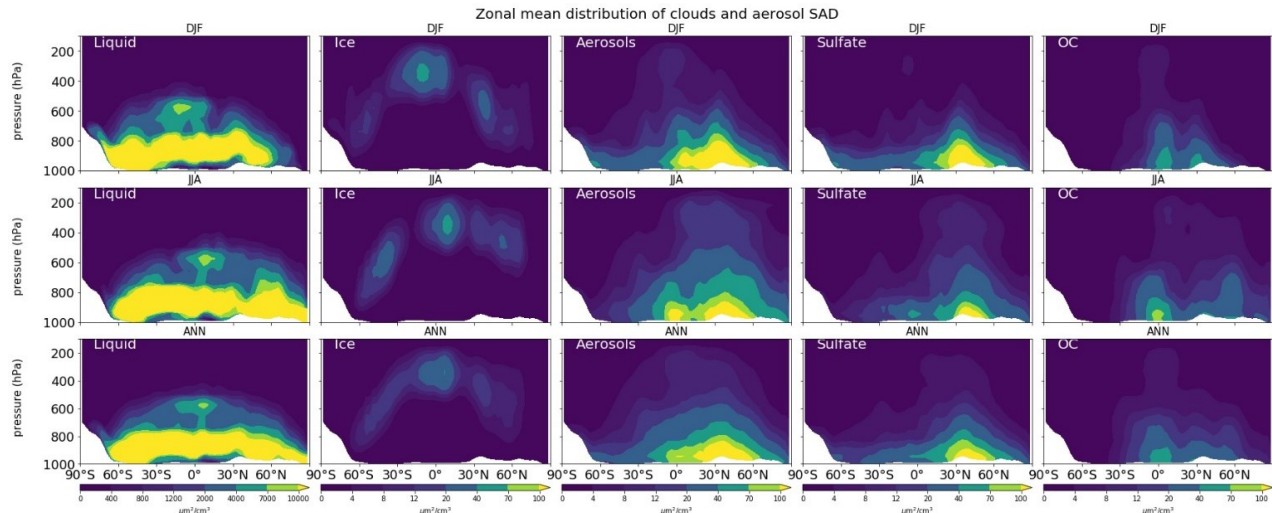

**Figure 8: Zonal, seasonal mean (upper and middle), and annual mean (lower) distribution of cloud droplet surface aerosol density**
**(SAD), cloud ice SAD, total aerosol SAD, sulfate aerosol SAD, and organic carbon SAD (from left to right).**



**Effects of N$_2$O$_5$ heterogeneous reaction (HR(N$_2$O$_5$))**

The inclusion of HR(N$_2$O$_5$) in the model increases global methane lifetime by +4.48% and changes NO$_x$, O$_3$ and CO abundances respectively by -5.51%, -2.12%, and +3.42% (Table 9).

In Fig. 9, the changes in OH, NO$_x$, O$_3$, and CO are most significant in the middle troposphere (400–600 hPa). These
changes are attributed mostly to uptakes of N$_2$O$_5$ onto aerosols, preferably onto clouds (apparent through correlation among effects by all HR(N$_2$O$_5$) and that by the HR(N$_2$O$_5$) onto total aerosols, Fig. 9). Marked negative effects on NO$_x$ concentration are apparent for DJF in the middle troposphere (600–700 hPa) of the 60° N and the Arctic region (>-20% at 700 hPa). The effects are probably associated with high concentrations of sulfate aerosols, organic carbons or soil dusts in the middle troposphere (see the paragraph above) and are also related to a long chemical lifetime of NO$_y$ in the middle-upper troposphere
in winter. When it comes to JJA, these negative effects become significant at higher altitudes around the 30°N/S (>-10% at 400 hPa). At the surface, HR(N$_2$O$_5$) causes negative effects on NO$_x$, O$_3$, OH concentrations (up to -23%, -4.5% and -7.5% respectively) and positive effects on CO concentration (up to +4.1%), also mainly attributable to the N$_2$O$_5$ uptake on aerosols.

In Fig. 10, the latitude–longitude means of HR(N$_2$O$_5$) effects are calculated for each pressure range (pressure ranges are defined as in Fig. 6). The global NO$_x$ decrease is up to -8.5% at 300–400 hPa. This decrease causes correspondent reductions
in O$_3$ and OH, which are calculated as about -3% and -7% at 400–600 hPa, respectively, for global mean O$_3$ and OH. About 3.5–4% global mean CO increment throughout the entire troposphere responds to decreased OH.

The small effects of HR(N$_2$O$_5$) on O$_3$ in the lower troposphere are consistent with findings from an earlier study (Riemer et al., 2003). Reductions in O$_3$ and NO$_x$ concentrations also well agree with the collective knowledge summarized in work reported by Brown and Stutz (2012). Despite a considerable HR(N$_2$O$_5$) effect calculated in the middle troposphere, its effect
in the whole troposphere is apparently not as great as reported to date. Another study assessed HR(N$_2$O$_5$) effects on annual burdens of NO$_x$, O$_3$, and OH, respectively as -11%, -5%, and -7% when using a similar $\gamma_{N2O5}$ value (0.1) (Macintyre and Evans, 2010). Although the effects of magnitude estimated in our work (Table 9) are almost half less than this earlier study (probably because of differences in NO$_x$ emissions, estimation of SAD, and chemical mechanism), the effect tendencies are similar. A strong increase of ozone attributed to N$_2$O$_5$ uptake under high-NO$_x$ conditions calculated using box models was
reported from an earlier study (Riemer et al., 2003), but this is only slightly apparent in our global model. Our results revealed that the HR(N$_2$O$_5$) effect might help clean up NO$_x$ pollutant. However, it increases the concentration of other pollutants (such as CO) because of the effects of reducing oxidizing agents in the atmosphere.



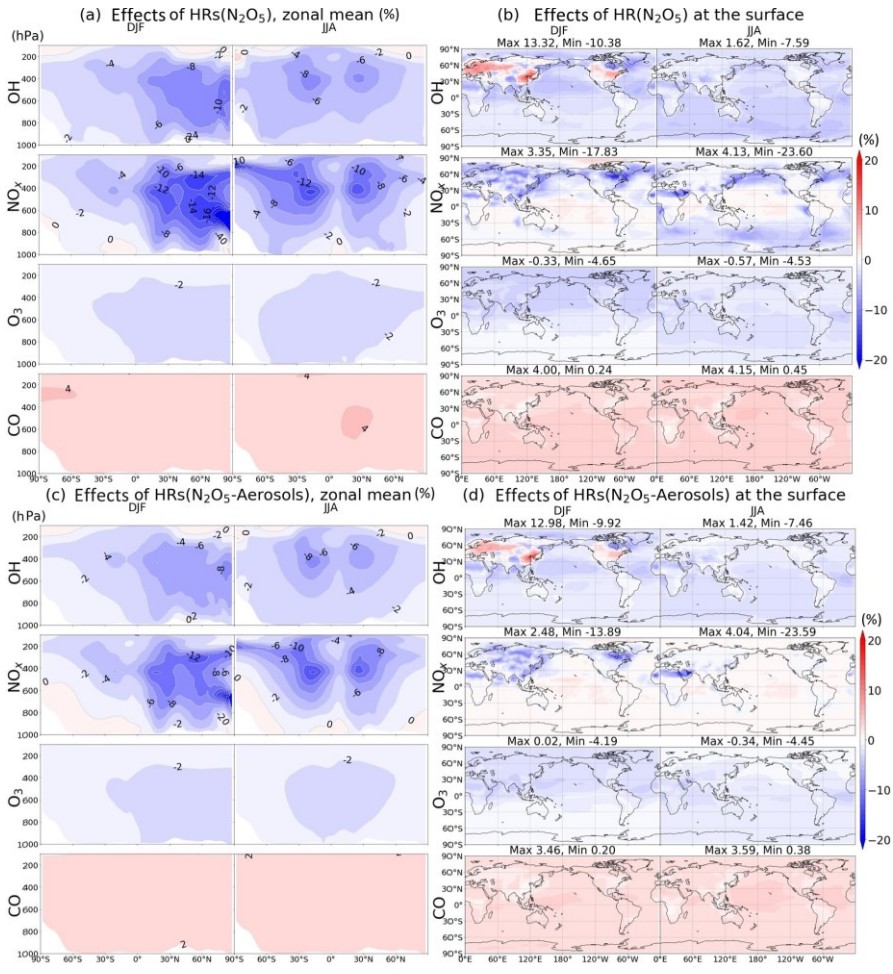

**Figure 9: Effects of HR(N₂O₅) in zonal-mean and at surface.**

a–b: Effects by $N_2O_5$ uptake onto both clouds and aerosols. c–d: Effects by $N_2O_5$ uptake onto aerosols.
Positive effects are presented in shades of red. Negative effects are in shades of blue. The contour interval is 2% in the plots for zonal mean.

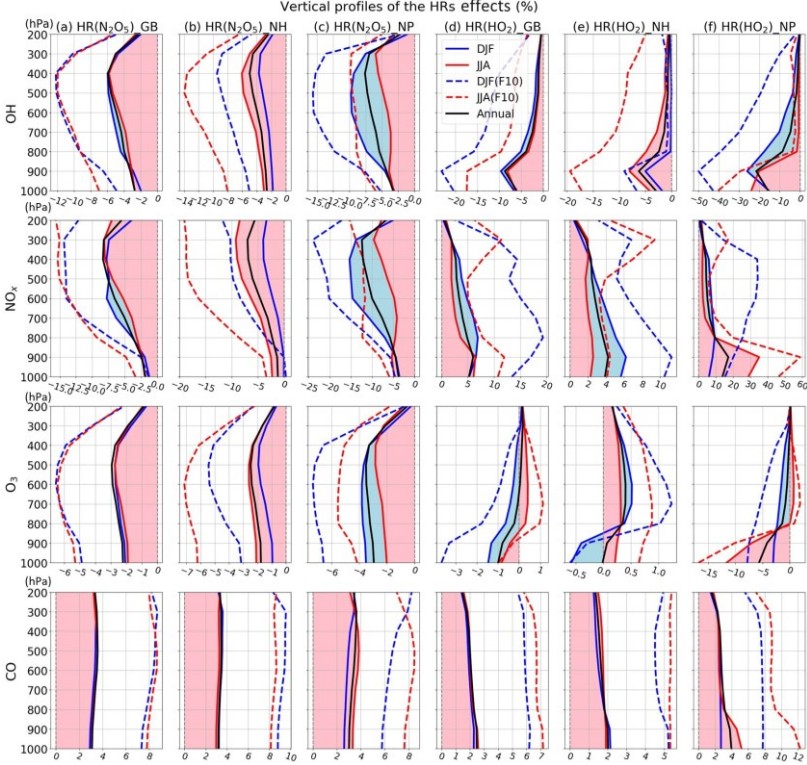

**Figure 10: Effects of N₂O₅ (a-c) and HO₂ uptakes (d–f) for each air pressure ranges. Calculations are for global (a,d), Northern Hemisphere (b,e), and North Pacific region (c,f). Additional dashed lines show effects by FCTHR_10 run (Sect. 3.3).**

**Effects of HO₂ heterogeneous reaction (HR(HO₂))**

Regarding the effects of HR(HO₂), the tropospheric methane lifetime increases by approx. 1.5%. Abundances in NO$_x$, O₃, and CO change, respectively, by +3.3%, +0.05%, and +1.9% (Table 9). In the entire troposphere, the influences of HR(HO₂) are not as large as that of HR(N₂O₅).

As Fig. 11 shows, the zonal-mean effects of HR(HO₂) on NO$_x$, OH, and O₃ are more widespread in DJF, but are more concentrated at the surface in JJA because of the high level of HO₂. The most substantial effects by HR(HO₂) are calculated in JJA at the surface of North Pacific (140–240° E, 40–60° N) by as much as +68.7% (NO$_x$), +7.29% (CO), -70% (OH), and -21% (O₃), which are more significant than those of HR(N₂O₅) at the surface. These effects are primarily attributable to HR(HO₂) in clouds rather than to aerosols (which is opposite to N₂O₅ uptake). These OH and O₃ reduction effects go along with past studies in which approx. 50% OH and approx. 10% O₃ reductions are calculated for the low troposphere of northern mid-latitude region ascribed to aqueous-phase HO$_x$ sink in clouds (Lelieveld and Crutzen, 1990, 1991). The efficient scavenge of HO₂ radical by cloud droplets might associate with acid–base dissociation HO₂/O₂⁻ and electron transfer of O₂⁻ to HO₂ to produce H₂O₂ (Jacob, 2000). Furthermore, cloud droplets SAD in our model are two orders of magnitude higher than total aerosol SAD (Fig. 8), which also contributes to the preference of the aqueous-phase HO₂ sink. Our large calculated effects for





the North Pacific region are new findings from other models, which have considered only aqueous aerosols (Stadtler et al., 2018; Thornton et al., 2008), because cloud particles are dominant at remote marine areas in addition to sulfate and aqueous sea salt particles (discussed at the beginning of Sect. 3.2). The $HO_2$ uptake onto aerosols is minor; it is observed only in DJF at the Arctic region and polluted areas (China and US), with apparent changes of up to +17% $NO_x$, -40% OH, and -14% $O_3$ at the local surface (panel b, Fig. 11). The aerosol negative-effect of HR($HO_2$) on surface $O_3$ concentration is significant at the

Chinese area, which might be in line with other studies of the Chinese $O_3$ trend (Kanaya et al., 2009; Li et al., 2019; Liu and Wang, 2020; Taketani et al., 2012), which suggests that the observed recent $O_3$ increases can be attributed to reduced $HO_2$ uptake under aerosol (PM) decreases brought about by the new Chinese Air Pollution policy.

       In Fig. 10, vertical profiles show that the latitude–longitude (lat-long) averaged effect of HR($HO_2$) on OH is -9% in the lower troposphere. As a result, the lat-long mean CO level increases +2.5% at the surface. Additionally, the daytime $NO_x$

oxidation by OH is suppressed. Also, $NO_x$ might be preserved in clouds (Dentener, 1993), which increases the lat-long averaged $NO_x$ level by +6% at 900 hPa. The lat-long mean $O_3$ is reduced by -1% at the surface, but it is increased at higher altitudes (about +0.2% at 300 hPa). The reduction of $O_3$ associates with $HO_2$ depletion in clouds and aqueous aerosols as described above, coupled with the $NO_x$ preservation in clouds, which enhance the $NO/NO_x$ ratio. The preserved $NO_x$ in clouds might remain available for $O_3$ production after the cloud evaporates (Dentener, 1993), along with the low SAD for both liquid

clouds and aerosols at higher altitudes (Fig. 8), thereby increasing $O_3$ in places other than aqueous phase. The $O_3$ increment might be trivial in DJF, but enhanced in JJA. As a result, the Northern Hemisphere-mean $O_3$ in JJA exhibits only positive effects. In contrast, for the North Pacific region in JJA, because of its large cloud fraction, an $O_3$ reduction effect is apparent in this region. The effects in JJA for this region show changes of -25% OH, +35% $NO_x$, -12% $O_3$, and +5% CO at 900–100 hPa as the most remarkable HR($HO_2$) effects as described above. In general, the regional mean effects of HR($HO_2$) in the

North Pacific region are enhanced in JJA, but the mean global effects of HR($HO_2$) are slightly favored in DJF because of the additional effects of aerosols during this season.

       Macintyre and Evans (2011) also found a similar contrast between the behaviors of HR($N_2O_5$) and HR($HO_2$): the uptake of $N_2O_5$ produces both regional and global effects on $O_3$, whereas the uptake of $HO_2$ affects $O_3$ at regional scales more strongly than on a global scale (Macintyre and Evans, 2011). Such features are generally consistent with our results.



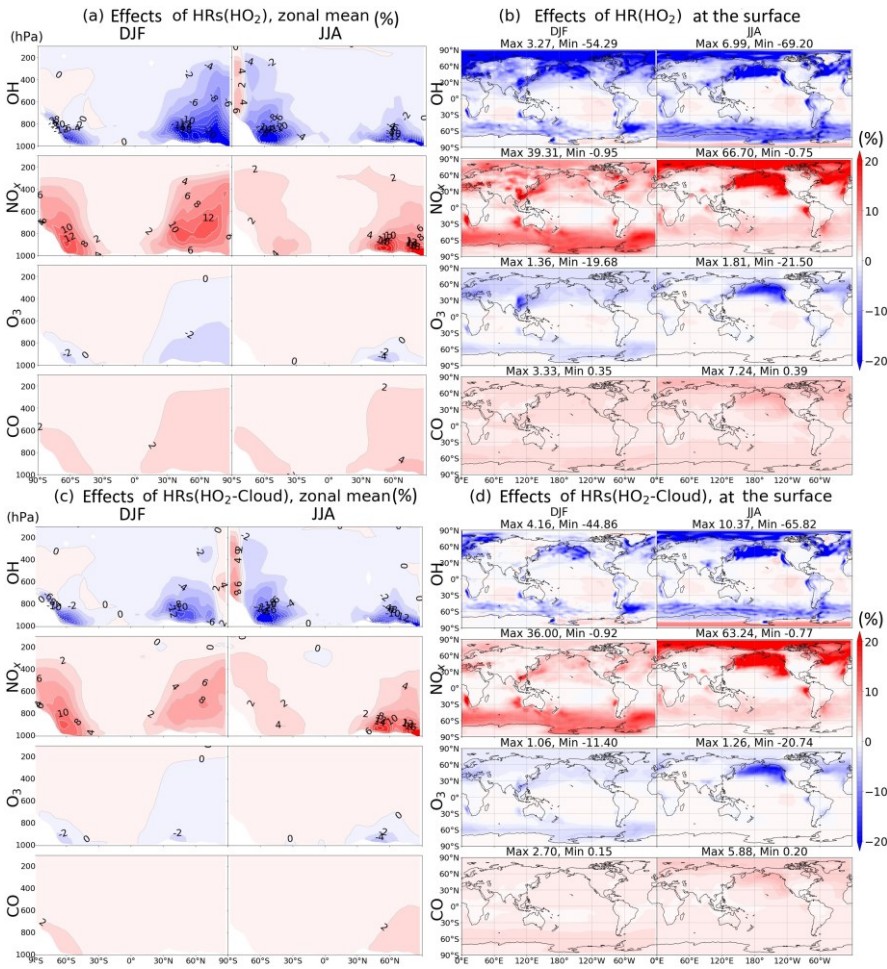

**Figure 11: Effects of HR(HO₂) in zonal mean and at the surface layer. a-b: Effects of HR(HO₂) onto both clouds and aerosols. c-d: Effects of HR(HO₂) onto clouds. The contour interval is 2% in the plots for zonal mean.**

**Effects of RO₂ heterogeneous reactions (HRs(RO₂))**

Effects of HR(RO₂) increase the global mean methane lifetime by +0.15% and change tropospheric abundances of NOₓ (+0.52%), O₃ (-0.93%), and CO (-1.78%) (Table 9). In Fig. 12, significant latitudinal contrasts exist in the NOₓ changes: large NOₓ increases at high latitudes with decreases at lower latitudes. These NOₓ changes probably reflect the reduced formation of PANs which decreases NOₓ transport from source regions to remote areas and from the surface to the upper troposphere (Villalta et al., 1996). The model calculated especially large NOₓ increases (>50%) for high latitudes around the Arctic sea in JJA, indicating reduction in the formation of PANs (NO₂ + RO₂ → PANs), which is linked tightly to the enhanced biogenic emissions of VOCs such as isoprene and terpenes in summer. The corresponding changes in OH concentration (because of the reduced NOₓ levels) at the surface are in the range of -4.6% to +20.4%. The effects of HR(RO₂) are primarily attributed to the heterogeneous reaction on clouds rather than on aerosols, although this cloud-effect is far smaller than the cloud-effect to the HO₂ uptake. Although it is proper to expect the high solubility of RO₂ (e.g. CH₃O₂) from its peroxy substituent (Betterton,





1992; Shepson et al., 1996), it is much less soluble than $HO_2$ because of its lower polarity, therefore the lower Henry law

constant (Jacob, 2000). Consequently, the possible accumulation of $CH_3O_2$ in the cloud is rather attributable to suppression of its gas-phase sink with $HO_2$ (Jacob, 1996).

Fig. 14 a–c show latitude–longitude means of $HR(RO_2)$ effects calculated for the respective pressure ranges: the latitude–longitude are constrained for the entire globe, the Northern Hemisphere, and North Pacific region. For the entire glob, the contrast effects of $HR(RO_2)$ between the lower and higher troposphere on $NO_x$ and OH are shown clearly (+3.5% $NO_x$ and

+0.55% OH at 900 hPa, but -2.5% $NO_x$ and -0.75% OH at 400–500 hPa annually). As a result, the annual mean $O_3$ and CO levels decreased throughout the troposphere, reaching their lowest at -1.6% $O_3$ and -1.5% CO at the surface. In JJA, the global effects by $HR(RO_2)$ are more concentrated in the lower troposphere, especially in the North Pacific (+3% OH, +10% $NO_x$, -3% $O_3$, -2% CO at 900–1000 hPa). In DJF, the $HR(RO_2)$ effects are observed mostly in the middle and higher troposphere, especially when considering the Northern Hemisphere (-1.25% OH, -4% $NO_x$, -2% $O_3$ at 500–800 hPa).

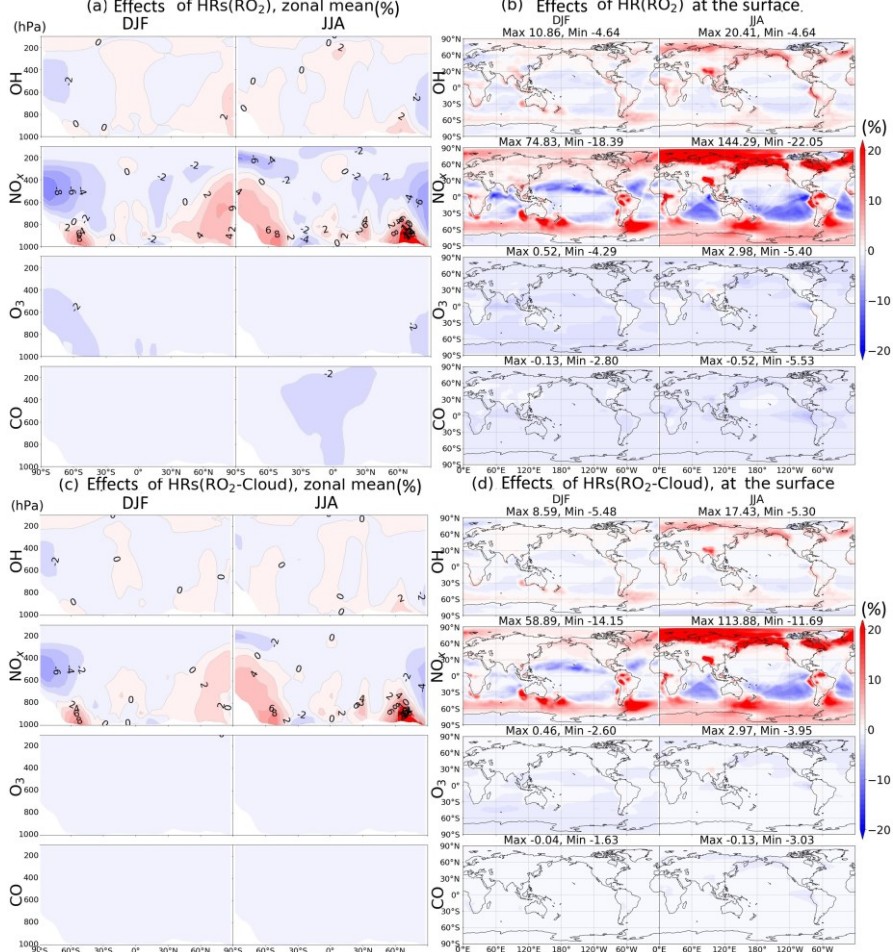


**Figure 12: Effects of HR(RO₂) in the zonal mean and at the surface layer. a-b show effects of the uptake onto both clouds and aerosols. c-d show separate effects of the uptake onto clouds. The contour interval is 2% in the plots for zonal mean.**



**Total effects of all HRs**

As discussed above, different heterogeneous reactions affect tropospheric chemistry differently. However, their effects can either augment or negate others in performing for the atmospheric chemistry. HR($N_2O_5$) is the greatest contributor to reduction of tropospheric OH, $O_3$, and $NO_x$ abundances, which is more active in the middle troposphere. HR($HO_2$) reduces OH, but increases the abundances of $O_3$ and $NO_x$ globally, whereas it exposes a negative effect on $O_3$ level at the surface of the North Pacific region. HR($RO_2$), similarly, has a smaller distribution to the total heterogeneous effects but its global-mean negative effects for $O_3$ are not negligible. The uptake of $N_2O_5$ mainly takes place to aerosols, whereas the uptakes of $HO_2$ and $RO_2$ occur more to liquid and ice clouds. Overall, the total effects of all HRs for the whole troposphere are +5.9% for global mean $CH_4$ lifetime, -2.2% for $NO_x$ (tropospheric abundance), -2.96% for $O_3$, and +3.3% for CO (Table 9). At the surface, the annual effects ranged from -52.7 to +2.3% for OH, -13.1 to +51.1% for $NO_x$, -13.1 to -1.5 for $O_3$, and -0.3 to +5.8% for CO (Fig. 13).

As Fig. 14 d–f show for the vertical profiles of HR effects, the change of OH largely concentrated in the lower troposphere (-10% OH at 900 hPa, calculated for the entire globe) is associated with the $HO_2$ uptake. By contrast, the $NO_x$ change is more intensive at higher altitudes (-9% $NO_x$ at 400 hPa, calculated for the entire globe), associated with $N_2O_5$ and $RO_2$ uptakes. The global-mean HR effects on $O_3$ and CO are vertically even, with the highest effects reaching -4.5% $O_3$ and +3.8% CO at the surface. Globally, HR effects on atmospheric oxidants (OH and $O_3$) are enhanced in DJF because of the higher pollution in the Northern Hemisphere. However, the largest HR effects are apparent for JJA at the surface of the North Pacific (-25% OH, +38% $NO_x$, -14% $O_3$, +6% CO as calculated for the 950–1000 hPa layer). These effects are mostly ascribed to $HO_2$ uptake onto clouds. This finding is also apparent from Fig. S15-c: these effects reach -66% for OH, +120% for $NO_x$, -23% for $O_3$, and +4.4% for CO at the surface. They were able to extend up to 400 hPa in the atmosphere. These substantial effects are readily apparent for the large reduction of $O_3$ level during MIRAI observation (red line versus green line in T5 bottom panel, Fig. 4). However, the major contribution of HR($HO_2$) to these effects is only partially verified by the ATom1 measurements in this study (red versus green lines in the bottom-right panel, Fig. 6). Because of model overestimates of cloud fraction in JJA for the North Pacific region, these effects of HR($HO_2$) should have existed at some smaller magnitude. For HR effects in the middle to upper troposphere, the $N_2O_5$ uptake on aerosols are mostly ascribed, which is intensive in both DJF and JJA.

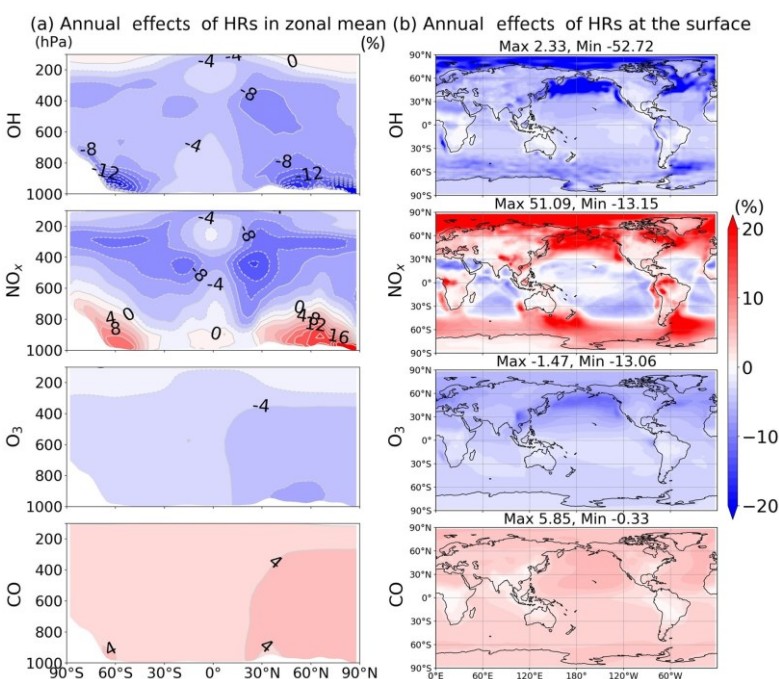

**Figure 13: Annual zonal-mean and surface total-HR effects.**

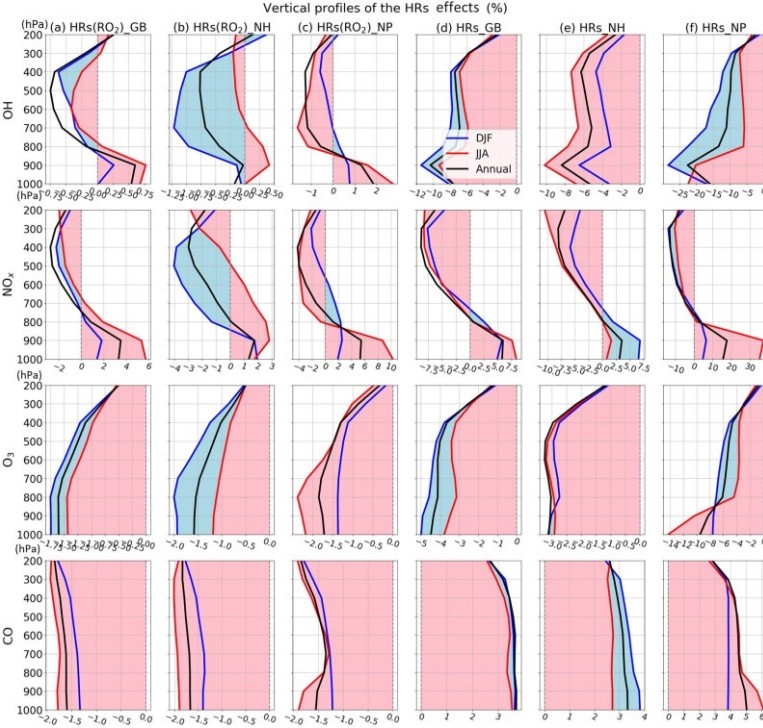


**Figure 14: Effects of RO₂ uptakes (a-c) and all HRs (d-f) averaged for each air pressure range. Calculations are for global (a,d), Northern hemisphere (b,e), and North Pacific region (c,f).**





**Table 9: Tropospheric abundances changes by HR(N$_2$O$_5$), HR(HO$_2$), HR(RO$_2$), and all HRs onto clouds and aerosols**

| | CH$_4$ lifetime (year) | Tropospheric abundances | | |
| --- | --- | --- | --- | --- |
| | | NO$_x$ (TgN) | O$_3$ (TgO$_3$) | CO (TgCO) |
| STD | 9.44 | 0.115 | 402.29 | 337.12 |
| noHR.n2o5 | 9.04 | 0.122 | 410.99 | 325.98 |
| noHR.n2o5-Cld | 9.41 | 0.116 | 402.81 | 336.43 |
| noHR.n2o5-Ae | 9.09 | 0.121 | 409.81 | 327.46 |
| noHR.ho2 | 9.30 | 0.111 | 402.09 | 330.67 |
| noHR.ho2-Cld | 9.35 | 0.113 | 402.25 | 332.97 |
| noHR.ho2-Ae | 9.40 | 0.114 | 402.17 | 335.29 |
| noHRs.ro2 | 9.43 | 0.114 | 406.06 | 343.23 |
| noHR.ro2-Cld | 9.42 | 0.115 | 403.96 | 339.06 |
| noHR.ro2-Ae | 9.45 | 0.115 | 404.03 | 340.76 |
| noHR | 8.91 | 0.118 | 414.55 | 326.43 |
| noHR.Cld | 9.32 | 0.113 | 404.55 | 335.03 |
| noHR.Ae | 9.06 | 0.119 | 411.38 | 329.20 |
| STD – noHR.n2o5 | +4.48% | -5.51% | -2.12% | +3.42% |
| STD – noHR.n2o5-Cld | +0.30% | -0.43% | -0.13% | +0.21% |
| STD – noHR.n2o5-Ae | +3.87% | -4.56% | -1.83% | +2.95% |
| STD – noHR.ho2 | +1.51% | +3.26% | +0.05% | +1.95% |
| STD – noHR.ho2-Ae | +0.41% | +1.11% | +0.03% | +0.55% |
| STD – noHR.ho2-Cld | +1.00% | +1.87% | +0.01% | +1.25% |
| STD – noHRs.ro2 | +0.15% | +0.52% | -0.93% | -1.78% |
| STD – noHRs.ro2-Cld | +0.23% | +0.39% | -0.41% | -0.57% |
| STD – noHRs.ro2-Ae | -0.12% | +0.09% | -0.43% | -1.07% |
| STD – noHR | +5.91% | -2.19% | -2.96% | +3.28% |
| STD – noHR.Cld | +1.34% | +1.71% | -0.56% | +0.63% |
| STD – noHR.Ae | +4.15% | -3.44% | -2.21% | +2.41% |






### 3.3 Sensitivities of tropospheric chemistry respond to heterogeneous reactions

From the discussion presented above, marked effects of HRs on global tropospheric chemistry are apparent. Here we examine how the tropospheric chemistry responds to the magnitude of HRs. To do this, we introduced a factor $F$ for application to the first-order loss rate shown in Eq. (1) for artificially manipulating the HR magnitude.

$$\beta_i = \sum_j (\frac{4}{v_i \gamma_{ij}} + \frac{R_j}{D_{ij}})^{-1} . A_j \times F \tag{8}$$

For this sensitivity test, we only specifically examine HR(HO$_2$) and HR(N$_2$O$_5$) and consider factors of 0–10 to the STD (Table S 1). This test might help to show the effective-oxidation sensitivity of the troposphere because future pollution and climate change might enhance the activities of these HRs.

For both effects, we performed nonlinear function fitting with their uptake loss rates, which yielded correlation coefficients
higher than 0.93 (Fig. 15). Although both HRs showed negative tendencies for OH and O$_3$ levels, the effect of HR(HO$_2$) on the tropospheric abundance of O$_3$ showed only a small increment with an increasing loss rate (maxima at around $F = 3$), and turned to reduction at higher rates ($F > 5$). As discussed along with HR(HO$_2$) effects, the O$_3$ level is expected to be reduced primarily only in JJA at the surface of the North Pacific region. At the same time, O$_3$ will be increased gradually elsewhere because of the persistent NO$_x$ increment. This behavior produces a positive global mean effect. Fig. 10 (dashed lines) shows
that manipulation of the HR(HO$_2$) loss rate ten factors higher will effectively increase the negative HR(HO$_2$) effects on O$_3$ in DJF (blue dashed versus solid blue lines, third row – fourth column panel), which results in a higher tendency of negative values for global-mean effects. This sensitivity in DJF might be attributable to the HO$_2$ uptake to aerosols rather than to clouds during this polluted period, which is apparent through comparison of Fig. 11 and Fig. S16 for notable events. In DJF, as amplifying a factor of 10 to HO$_2$ uptake loss rate, the effects for the polluted Chinese area (because of HO$_2$ uptake onto
aerosols) significantly magnify from -18% (third row – first column in panel b, Fig. 11) to -47% (third row – first column in panel b, Fig. S16). In contrast, effects at the surface O$_3$ level in JJA for the North Pacific region (because of HO$_2$ uptake onto clouds) only enhance from -21% (third row-second column in panel b, Fig. 11) to -29% (third row – second column in panel b, Fig. S16).

As amplifying a factor of 10 to HR(N$_2$O$_5$), the sensitivities of global effects show no seasonal variation. The HR(N$_2$O$_5$)
effects are more sensitive in DJF for the North Pacific region, which link to the higher concentration of aerosol in this season. Otherwise, the HR(N$_2$O$_5$) effects for the generic Northern Hemisphere tend to be more sensitive in JJA as a result of pollutant transportation to the higher troposphere.

Consequently, we suggest that the sensitivity of tropospheric chemistry to HR(N$_2$O$_5$) and HR(HO$_2$) might be attributable to loss activities to aerosols rather than to clouds. The sharp-curved effect on O$_3$ because of amplification of HR(HO$_2$) makes
sense in plans for ozone pollution control when increased pollution or climate change factors cause the rate of HRs in the future to increase by 3–5 times or more.



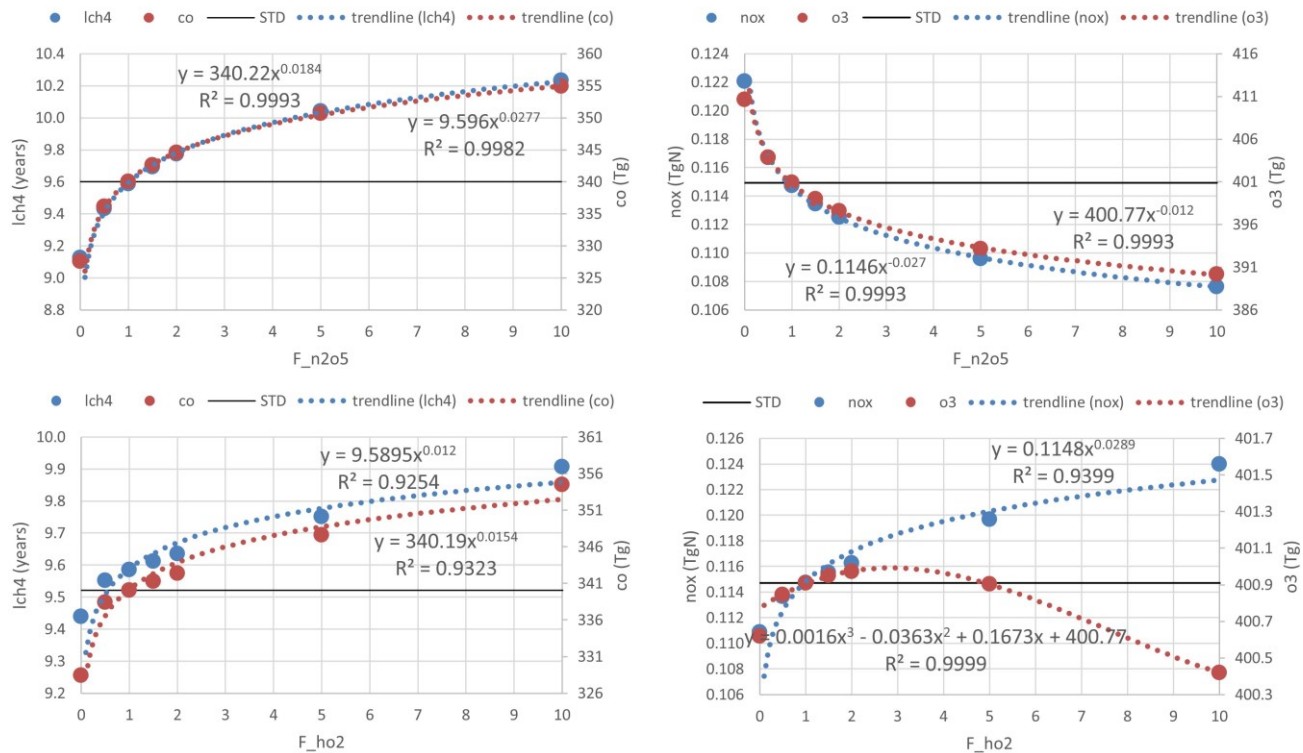

**Figure 15: Trendlines for the sensitivity of HR(N₂O₅) (upper panel) and HR(HO₂) effects (lower panel) with uptake rates. The left panels show the CH₄ lifetime (blue) and tropospheric abundance of CO (red). The right panels show tropospheric abundances of NOₓ (blue) and O₃ (red).**

## 4 Conclusion

The "CHASER" chemistry–climate model was used to investigate global effects of $N_2O_5$, $HO_2$, and $RO_2$ uptake. Verification of the model with observations from inland and ocean domains showed adequate agreement for $PM_{2.5}$, $SO_4^{2-}$, $NO_3^-$ particles, gaseous $HNO_3$, $NO_x$, OH, CO, and $O_3$ concentrations. $R$, bias, and NRMSE values for $SO_4^{2-}$, $NO_3^-$, and $HNO_3$ at EANET and EMEP stations are comparable with other models. Inclusion of HR reduced model bias for OH, $NO_2$, CO, and $O_3$, especially in the low troposphere. However, verification with satellite and reanalysis data showed deterioration by HRs for TCO, and an overestimate for cloud fraction in the North Pacific region.

The total effects of HRs are important for the tropospheric chemistry that might change +5.9% $CH_4$ lifetime, -2.19% $NO_x$, -2.96% $O_3$, and +3.3% CO abundances. Global effects are -9% $NO_x$ at 400 hPa, -10% OH at 900 hPa, -4.5% $O_3$ and +3.8% CO at the surface. Global HR effects tend to be enhanced in DJF because of greater amounts of pollution in the Northern Hemisphere.

Total HR effects are contributed mainly by HR($N_2O_5$) onto aerosols in the middle troposphere. At the surface, HR($HO_2$) is more active and leaves a remarkable disturbance in JJA at the North Pacific region with changes of -70% for OH, -24% for $O_3$, +68% for $NO_x$, and +8% for CO. These effects were attributed to the uptake of $HO_2$ on cloud particles, which were partially





verified with ATOM1 observations. However, the effect magnitude requires further investigation because of model overestimates for cloud fractions in this region.

The sensitivity of tropospheric chemistry with the HR magnitude was determined as nonlinear functions. The increasing effect for the global $O_3$ abundance by HR($HO_2$) will sharply change to a decreasing effect when the uptake rate is amplified by more than three times. This turning is ascribed to the uptake onto aerosols in DJF. In general, uptake to aerosols is more

responsive to the heterogeneous loss rate than uptake to clouds.

Overall, the $N_2O_5$ and $HO_2$ uptakes will sweep away atmospheric oxidants, thereby enhancing concentrations of pollutants. Our results reveal that although HRs are reported to be associated with polluted regions, the global effects of HRs reach further remote regions such as the marine boundary layer at the middle latitude and the upper troposphere. For ground-based studies of polluted regions such as China, it should be considered that HR($HO_2$) and HR($RO_2$) were able to contribute respectively to

the $NO_x$ increment in DJF and JJA. Moreover, the HR($HO_2$) effect might hinder efforts at reducing environmental pollution in urban areas because it increases $NO_x$ but decreases $O_3$ at the surface. Therefore, if this reaction is minimized because of a decrease in particulate matter, then the surface ozone level might increase.

**Code availability**

The source code for CHASER V4.0 and input data to reproduce results in this work can be obtained from the repository at

http://doi.org/10.5281/zenodo.4153452 (Ha et al., 2020).

**Author contribution**

Ha T.M.P. performed all simulations (except simulations for the cloud-fraction validation), interpreted the results and wrote the manuscript. Sudo K. developed the model code, conceived of the presented idea, supervised the findings of this work and the manuscript preparation. Matsuda R. carried out the simulations and plots for the validation of cloud fraction. Kanaya Y.

and Taketani F. provided the R / V MIRAI ship data as well as contributed to the discussion of the work's findings.

**Competing interests**

The authors declare that they have no conflict of interest.

**Acknowledgments**

This research was supported by the Global Environment Research Fund (S–12) of the Ministry of the Environment (MOE),

Japan, and by JSPS KAKENHI Grant Numbers: JP20H04320, JP19H05669, and JP19H04235. We are grateful to the NASA scientists and staffs for providing ATom data (https://espo.nasa.gov/atom/content/ATom). The simulations were completed





using the supercomputer (NEC SX-Ace and SX-Aurora TSUBASA) at NIES, Japan. The surface observational data for the model validation were taken from the monitoring networks EANET (https://www.eanet.asia/) and EMEP (https://www.emep.int/).

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
