# Peer review of "Effects of heterogeneous reactions on tropospheric chemistry: a global simulation with the chemistry climate model CHASER V4.0"

_Geoscientific Model Development, 2020_

## Short Comment (SC1) · 14 Nov 2020

Dear authors,

in my role as Executive editor of GMD, I would like to bring to your attention our Editorial version 1.2:

https://www.geosci-model-dev.net/12/2215/2019/

This highlights some requirements of papers published in GMD, which is also available on the GMD website in the 'Manuscript Types' section:

http://www.geoscientific-model-development.net/submission/manuscript_types.html

In particular, please note that for your paper, the following requirements have not been

met in the Discussions paper:

- "The main paper must give the model name and version number (or other unique identifier) in the title."

- "If the model development relates to a single model then the model name and the version number must be included in the title of the paper. If the main intention of an article is to make a general (i.e. model independent) statement about the usefulness of a new development, but the usefulness is shown with the help of one specific model, the model name and version number must be stated in the title. The title could have a form such as, "Title outlining amazing generic advance: a case study with Model XXX (version Y)"."

Therefore please add the model name and version number (CHASER v4.0) in a similar way as proposed to the title of your manuscript upon revised submission.

Yours,

Astrid Kerkweg

---

## Author Comment (AC1) · 17 Nov 2020

Response to Astrid Kerkweg's short comment,

We thank Astrid Kerkweg as Executive editor of GMD for bringing our attention to the Editorial version 1.2 which relates to the model name and version number required in the title of our paper.

For that, we are pleased to add the model name and version number (CHASER V4.0) to the title of our manuscript upon the revised submission. If there are no other upcoming comments relates to the title, the new title is intended to be: "Effects of heterogeneous reactions on global tropospheric chemistry: a case study with model CHASER V4.0"

We hope this revision will comply with GMD requirements.

[Figure]

Sincerely,

On behalf of all co-authors, Phuc T. M. Ha.
* * *

---

## Referee Comment (RC1) · Anonymous Referee #1 · 28 Dec 2020

This paper evaluated the effects of heterogeneous uptake reactions of $N_2O_5$, $HO_2$ and $RO_2$ on cloud and aerosol particles by using a chemical-climate model CHASER, and the modelling results have been verified by comparing with ground-based measurements, shipboard, aircraft and satellite observations. Although the findings of this study on the changes in global abundances of $NO_2$, $NO_3$, $O_3$, and CO, and lifetime of $CH_4$ are basically within the range of uncertainties of previous studies, and no new surprising finding are reported, this work provides the most comprehensive view among this kind of studies covering the lower to upper troposphere, polluted terrestrial and remote oceanic region, and seasonal to annual characteristics. Particularly the study demonstrated the heterogeneous effect in the remote areas such as oceanic region

and the upper troposphere for the first time. Tthe present reviewer judges this paper is acceptable for publication after considering the following comments.

1. The difference between the role of uptake of HO2 and RO2 should be explained more in detail. In the case of the uptake of RO2, the reduction of the formation of PAN and organic nitrates due to the reactions, CH3COO2 + NO2 ïĆő PAN, and RO2 + NO ïĆő RONO2, as well as the reduction of NO oxidation reaction, RO2 + NO ïĆő RO + NO2, RO + O2 ïĆő HO2 etc. are expected. How the difference in the effect of HR(HO2) and HR(RO2) shown in Figs. 11 and 12 can be explained by these factors? 2. Other than the well-known heterogenous processed of N2O5, HO2 and RO2 analyzed in this study, the heterogeneous renoxification process of HNO3 to reproduce NOx has previously been suggested in order to explain the model overestimate of HNO3/NOx ratio in the free and polluted atmosphere (Hauglustaine et al., Geophys. Res. Lett., 23, 2609-2612, 1996: Lary et al., J. Geophys. Res., 102, 3671–3682, 1997; Li et al., SOLA, 11, 124–128, 2015; Akimoto et al., Atmos. Chem. Phys., 19, 603-615, 2019). Although the importance of this process has not been established, the same tendency of overestimate of HNO3 and underestimate of NOx has been revealed in this study (Table 5). Discussion should be given for the possibility of the heterogeneous reaction of HNO3 whether in supporting or objecting. 3. Many of the figures are rather poorly presented for readers and should be revised. (1) In most of the figures, size of inside letters and axis labels are too small (unreadable on print and difficult to read even on PC screen). (2) Fig.3: How the site for each species were selected? There is no explanation in the text. (3) Figs. 3, 4, 5: The difference between the plots for noHR_n2o5, _ho2, _ro2 and _CLD are almost undiscernible. It is suggested to show only noHR and STD in these Figures, and the difference of noHR_n2o5, _ho2, _ro2 and _CLD should be presented in some selected plots in a different Figure. (4) Figs. 9, 11, 12: The differences between the upper and lower figures are not discernible easily. It is suggested to delete the figures for HRS(N2O5-aerosols), HRS(HO2-Cloud) and HRS(RO2-Cloud) in these Figures. It would be enough to explain in the text that the uptake of N2O5 on aerosols, and that of HO2 and RO2 on cloud are major processes.

Explanation should be given in the text why the process predominate for each of the species. (5) Figs. 10, 14: Labels and units of horizontal axis should be given properly. 4. Table 2: What is the meaning of asterisk for "product*". What do the ISO2 and MACRO2 stand for? 5. Tables 5, 6, 7, 8: Units should be given appropriately.

over
* * *

---

## Referee Comment (RC2) · Anonymous Referee #2 · 9 Apr 2021

This manuscript presents a global model-based study of the impacts of various heterogeneous uptake processes on tropospheric composition. The CHASER model is validated against a number of ground-, ship-, aircraft-, and satellite-based observations of relevant species ($NO_x$, PM, $O_3$, CO, OH, and total column $O_3$) and parameters (cloud fraction). A variety of simulations are designed to probe aerosol, cloud droplet, and ice particle uptake of $N_2O_5$, $HO_2$, and $RO_2$, with individual species and uptake pathways (cloud versus aerosol) turned off in turn. Total and spatially-/temporally-resolved changes in methane lifetime, $NO_x$, $O_3$, and CO are assessed. Finally, a sensitivity simulation is conducted to evaluate the impacts to atmospheric composition with variations in the examined heterogeneous loss rates.

The study presents a well-rounded analysis of heterogeneous uptake from a global model perspective. The model is thoroughly assessed against a reasonable number of available observations. The thoroughness of sensitivity simulations, both turning on/off all and individual heterogeneous uptake reactions as well as varying the magnitude of the first-order loss rate applied, addresses in a methodical way how this chemistry might impact global composition. I think there may be a missed opportunity in this manuscript to more thoroughly discuss the mechanisms through which these impacts manifest, but diagnostics necessary to perform such assessments may be lacking, and this should not preclude publication of this work. There is also a concern that model biases (such as the overestimation in cloud fraction in the northern Pacific lower troposphere) will introduce model-dependent errors in the results, though I regard the acknowledgment of this issue in the text as sufficient. I would consider this work as suitable for publication in this journal following the incorporation of the suggestions noted below, primarily concerned with clarifications and organization.

Major Comments

In addition to collective improvements to figure clarity and organization, noted below under minor comments, my only other major request would be to expand on discussions of the mechanisms underlying some of the changes exhibited in the presented results. For instance, the description of effects on PAN production and transport, ∼L492, could be more explicit and re-emphasized in the Conclusions. The reasons for the large increases in NOx near the surface in the Arctic during JJA due to HR(RO2) are still unclear to me – if there were a large source of NOx here, the reduction in PAN formation may make sense, but large sources at these high latitudes seems unlikely. Similarly, why are there increases in NOx during DJF due to HR(RO2) in the high latitude southern oceans, just offshore?

Similarly, one impact attributed to RO2 uptake is a decrease in CO (e.g., L257). I'm curious about the mechanism, and not aware of any discussion regarding this. I would assume that the CO decrease is due simply to reductions in secondary production –
functionalized C-containing RO2 species that would otherwise be oxidized to form CO are instead taken up on aerosols/cloud droplets. Do the authors know if this is the case, or if there is another mechanism at play?

I understand that diagnosing these kinds of questions from global model output, especially from lengthy and numerous simulations, may be difficult, given limitations on how much output can be generated. While model evidence to further describe these mechanistic questions would be ideal, hypotheses supported from prior literature or simpler, logical arguments would suffice.

Minor Comments

L120: Please add a reference for the MAC reanalysis biomass burning emissions, or else provide more detail

~L175: Somewhere in Section 2, the timeframe of the simulations should be clearly stated. I gather from some of the time series figures that model output is available for at least 2010-2018 – were all sensitivity simulations run for this entire period?

~L255: The discussion surrounding Fig. 3 refers to differences between the various model sensitivity runs, but it is very difficult to make out the different colored lines representing the different simulations in the figure. Perhaps an inset that shows a "zoomed in" view of a representative portion of each panel, or else plotting in different coordinates, like % difference compared to obs versus time, would help remedy the issue. This applies to Figures 4 and 5 as well.

L286: The suggestion of insufficient downward mixing of stratospheric air in the model while CO is underestimated by the model seems counterintuitive to me. Stratospheric air should be depleted in CO, so I would expect higher observed CO would point to something other than stratospheric influence, especially at the surface, as the ship-based observations are. I'm unfamiliar with Kanaya et al.; do they provide some other rationale to explain this apparent discrepancy?

L334: It is unclear to me why the "ground layer" is defined differently for all flights (> 800 hPa) versus for the N. Pacific region (> 700 hPa) in Table 8 – could the authors include a brief explanation?

L365: I feel that the TCO plots in Fig. 7 would be more easily understood by plotting Model – OMI differences, for both the STD and noHR simulations. As is, the differences between the model runs and OMI stand out far more than the differences between the STD and noHR runs. One really has to focus on small details to see where the model is improving with respect to OMI.

L466: The statement that "recent O3 increases can be attributed to reduced HO2 uptake under aerosol (PM) decreases brought about by the new Chinese Air Pollution policy," is, I think, too strongly worded without a quantitative accounting of the observed O3 changes. Other effects, such as the non-linearity in O3 production with NOx concentration, could also be contributing. Qualifying the statement as "can be attributed in part to reduced HO2. . ." or similar, would be sufficient.

L548: "magnitude of HRs" is vague; could you clarify if this sensitivity test is meant to probe uncertainties in the first-order loss rate, possible non-linearities in the uptake, etc.?

L589: I'd suggest staying consistent with the number of significant figures reported in the % changes here, in the abstract (L13), and elsewhere. Sometimes one digit is reported after the decimal place, sometimes two, and sometimes none.

Figs. 10 and 14: x-axis labels would be helpful, for anyone who may miss the (%) in the title.

Technical corrections

L130: "uncertainties" should be "uncertain"

L293: "undervalues" is a slightly out-of-place word choice; "underestimates" may be better

L297: The use of "extends" here suggests that model underestimates are extending in time/space instead of getting worse. "worsens" or "exaggerates" may better reflect the intended meaning.

L301: A verb is needed in this sentence; "Ocean is mostly dominated. . ."

L325: "However, for the. . ." the "for" is not needed.

L326: Here and elsewhere, I'd suggest the authors check for consistency in how "ATom-1" is capitalized and punctuated.

L364: The order of appearance of the Supplemental figures should match the order in which they are mentioned in the main text.

L380: "surface aerosol density" should be "surface area density"

L415: "preferably onto" confuses the meaning of this sentence; I suggest "rather than onto" to emphasize the importance of aerosol uptake over cloud uptake

L503: "glob" should be "globe"

L502: Fig. 14 is introduced here before Fig. 13 is discussed; I'd suggest switching the two.

L536: The phrase "$N_2O_5$ uptake on aerosols are mostly ascribed" would be more easily understood as "$N_2O_5$ uptake is mostly ascribed to aerosols"

---

## Author Comment (AC3) · 14 May 2021

Response to Anonymous Referee #1's comment, We thank the Anonymous Referee for the thorough comments on our manuscript.

Referee's comment: This paper evaluated the effects of heterogeneous uptake reactions of N2O5, HO2 and RO2 on cloud and aerosol particles by using a chemical-climate model CHASER, and the modelling results have been verified by comparing with ground-based measurements, shipboard, aircraft and satellite observations. Although the findings of this study on the changes in global abundances of NO2, NO3, O3, and CO, and lifetime of CH4 are basically within the range of uncertainties of previous studies, and no new surprising finding are reported, this work provides the

most comprehensive view among this kind of studies covering the lower to upper tro-
posphere, polluted terrestrial and remote oceanic region, and seasonal to annual char-
acteristics. Particularly the study demonstrated the heterogeneous effect in the remote
areas such as oceanic region and the upper troposphere for the first time. Tthe present
reviewer judges this paper is acceptable for publication after considering the following
comments.

Author's response: We genuinely appreciated the productive comments from Referee
**1 for our work.**

Referee's comment: 1. The difference between the role of uptake of HO2 and RO2
should be explained more in detail. In the case of the uptake of RO2, the reduction
of the formation of PAN and organic nitrates due to the reactions, CH3COO2 + NO2
→ PAN, and RO2 + NO → RONO2, as well as the reduction of NO oxidation reaction,
RO2 + NO → RO + NO2, RO + O2 → HO2 etc. are expected. How the difference in
the effect of HR(HO2) and HR(RO2) shown in Figs. 11 and 12 can be explained by
these factors?

Author's response: We added an explanation for better clarifying the difference be-
tween the role of HO2 and RO2 uptakes. Both HR(HO2) and HRs(RO2) suppress
the NO oxidation, which is respectively via reactions (R1) and (R2-R3-R1): HO2 +
NO → OH + NO2 (R1) RO2 + NO → RO + NO2 (R2) RO + O2 → R'O + HO2 (R3)
Thus, the uptakes of HO2 and RO2 both preserve high NO/NOx ratio and generally
restrict OH and O3 formations (Fig. 11 i, j, m, n). However, less RO2 participating
in the hydrocarbon oxidation only reduces OH and O3 levels at polluted region while
enhances OH level and leave no significant effect on O3 at remote regions (Fig. 12 i, j,
m, n), due to the different oxidizing mechanisms for HCs between polluted and remote
regions. Moreover, HRs(RO2) do suppress the formations of PAN and other organic
nitrates. Less PAN is produced, which means more NOx are preserved, esp. at the
lower troposphere. Fig.12 l showed a doubly maximum increase for NOx at the sur-
face (144%) compared to the maximum NOx increase seen in Fig. 11 l (66%), due to

reducing effects by HRs(RO2) for both NO oxidation and PAN formation. We provided the additional explanation for our manuscript at L543-569.

Referee's comment: 2. Other than the well-known heterogenous processed of N2O5, HO2 and RO2 analyzed in this study, the heterogeneous renoxification process of HNO3 to reproduce NOx has previously been suggested in order to explain the model overestimate of HNO3/NOx ratio in the free and polluted atmosphere (Hauglustaine et al., Geophys. Res. Lett., 23, 2609-2612, 1996: Lary et al., J. Geophys. Res., 102, 3671–3682, 1997; Li et al., SOLA, 11, 124–128, 2015; Akimoto et al., Atmos. Chem. Phys., 19, 603-615, 2019). Although the importance of this process has not been established, the same tendency of overestimate of HNO3 and underestimate of NOx has been revealed in this study (Table 5). Discussion should be given for the possibility of the heterogeneous reaction of HNO3 whether in supporting or objecting.

Author's response: We thank for the suggestion from the Referee. In the original text, the comparison with ground observations for EANET and EMEP still showed low correlations for HNO3 (0.177 for EANET, 0.116 for EMEP – Table 5). In Fig. S9, the model correlations with EANET and EMEP observations for HNO3 and NOx showed higher tendencies for HNO3 overestimates and NOx underestimates at low levels of HNO3 (0-1 ppb) and high levels of NOx (>10 ppb), which indicate the highly polluted sites. For remote regions covered by ATom1 flights, our model showed relatively large overestimates for NOx at the surface layer (Fig. 6 a,e). The low reproducibility of model for NOx could be due to the low horizontal resolution of the simulations ($\sim$ 2.8o). Higher resolutions could improve the model reproduction for surface NOx as previously investigated by Sekiya et al. (Geosci. Model Dev., 11, 959–988, 2018).

The heterogeneous "renoxification" reaction of HNO3 on soot surface (R4), which is suggested by the Referee, could also be a possible solution: HNO3 + soot $\rightarrow$ NO + NO2 (R4). The additional (R4) followed by NO2 uptakes onto soot: NO2 + particles $\rightarrow$ 0.5 HONO + 0.5 HNO3 (R5), can be expected to increase NO, and decrease O3 via the consequent titration reaction. These changes could reduce the model overestimates

for HNO3 and O3, and the model underestimates for NOx with EANET and EMEP stations.

A concerning NOx chemistry regarding HONO formation is already considered in another report (preparing for submission), coupling several HONO reactions including (R5). Without (R4)'s inclusion, the whole HONO chemistry could either increase or decrease HNO3 at EMEP and EANET stations during winter and summer conditions, resulting in slight reductions for model bias with EANET and EMEP for HNO3. However, the comparison with ground observations for NOx was not improved. When we incorporate (R4) into the model, the NOx chemistry did not undergo an effective "renoxification" to enhance NOx concentrations over EANET. To be able to conclude whether the "renoxification" process could remedy the issue, further examination would be required.

We revised our manuscript based on the above explanation at L267-276.

Referee's comment: 3. Many of the figures are rather poorly presented for readers and should be revised. (1) In most of the figures, size of inside letters and axis labels are too small (unreadable on print and difficult to read even on PC screen). (2) Fig.3: How the site for each species were selected? There is no explanation in the text. (3) Figs. 3, 4, 5: The difference between the plots for noHR_n2o5, _ho2, _ro2 and _CLD are almost undiscernible. It is suggested to show only noHR and STD in these Figures, and the difference of noHR_n2o5, _ho2, _ro2 and _CLD should be presented in some selected plots in a different Figure. (4) Figs. 9, 11, 12: The differences between the upper and lower figures are not discernible easily. It is suggested to delete the figures for HRS(N2O5-aerosols), HRS(HO2-Cloud) and HRS(RO2-Cloud) in these Figures. It would be enough to explain in the text that the uptake of N2O5 on aerosols, and that of HO2 and RO2 on cloud are major processes. Explanation should be given in the text why the process predominate for each of the species. (5) Figs. 10, 14: Labels and units of horizontal axis should be given properly.

Author's response: (1), (5) We acknowledged the responsibility for the figures' readability. We considerably modify each figure in the revised version upon the Referee's suggestions. (2) We revised Fig. 3 with presenting the median value of grouped stations as Chinese region (stations in China and South Korea), remote stations with low NOx levels of EANET, and all EMEP stations. (3) In Fig. 3,4,5, we separate plots of each HR impacts from the concentration plots, at which we keep only noHR and STD's comparison with measurements. (4) Figures for HRS(N2O5-aerosols), HRS(HO2-Cloud) and HRS(RO2-Cloud) are moved to the Supplement.

Referee's comment: 4. Table 2: What is the meaning of asterisk for "product*". What do the ISO2 and MACRO2 stand for?

Author's response: The asterisk for "product*" in Table 2 was meant to represents a remaining error of expression and was deleted in the revised version. ISO2 denoted for peroxy radicals from C5H8+OH, and MACRO2 stands for peroxy radicals from the oxidation of MACR, methacrolein (CH2=C(CH3)CHO). These descriptions were provided in the revised version at L194-195.

Referee's comment: 5. Tables 5, 6, 7, 8: Units should be given appropriately.

Author's response: We provided the units in these tables appropriately in the revised version.

Sincerely,

On behalf of all co-authors, Phuc T. M. Ha.

---

## Author Response (AR1)

Response to the Anonymous Referee #1's comment,

We thank the Anonymous Referee for the thorough comments on our manuscript.

**Referee's comment:** This paper evaluated the effects of heterogeneous uptake reactions of N2O5, HO2 and RO2 on cloud and aerosol particles by using a chemical-climate model CHASER, and the modelling results have been verified by comparing with ground-based measurements, shipboard, aircraft and satellite observations. Although the findings of this study on the changes in global abundances of NO2, NO3, O3, and CO, and lifetime of CH4 are basically within the range of uncertainties of previous studies, and no new surprising finding are reported, this work provides the most comprehensive view among this kind of studies covering the lower to upper troposphere, polluted terrestrial and remote oceanic region, and seasonal to annual characteristics. Particularly the study demonstrated the heterogeneous effect in the remote areas such as oceanic region and the upper troposphere for the first time. Tthe present reviewer judges this paper is acceptable for publication after considering the following comments.

**Author's response:** We genuinely appreciated the productive comments from the Referee #1 for our work.

**Referee's comment: 1.** The difference between the role of uptake of HO2 and RO2 should be explained more in detail. In the case of the uptake of RO2, the reduction of the formation of PAN and organic nitrates due to the reactions, CH3COO2 + NO2 → PAN, and RO2 + NO → RONO2, as well as the reduction of NO oxidation reaction, RO2 + NO → RO + NO2, RO + O2 → HO2 etc. are expected. How the difference in the effect of HR(HO2) and HR(RO2) shown in Figs. 11 and 12 can be explained by these factors?

**Author's response:** We added an additional explanation for better clarifying the difference between the role of $HO_2$ and $RO_2$ uptakes. Both $HR(HO_2)$ and $HRs(RO_2)$ suppress the NO oxidation, which is respectively via reactions (R1) and (R2-R3-R1):

$HO_2 + NO →$ $OH + NO_2$ (R1)

$RO_2 + NO →$ $RO + NO_2$ (R2)

$RO + O_2 →$ $R'O + HO_2$ (R3)

Thus, the uptakes of $HO_2$ and $RO_2$ both preserve high $NO/NO_x$ ratio and generally restrict OH and $O_3$ formations (Fig. 11 i, j, m, n). However, less $RO_2$ participating in the hydrocarbon oxidation only reduces OH and $O_3$ levels at polluted region while enhances OH level and leave no significant effect on $O_3$ at remote regions (Fig. 12 i, j, m, n), due to the different oxidizing mechanisms for HCs between polluted and remote regions.

Moreover, $HRs(RO_2)$ do suppress the formations of PAN and other organic nitrates. Less PAN is produced, which means more $NO_x$ are preserved, esp. at the lower troposphere. Fig.12 l showed a doubly maximum increase for $NO_x$ at the surface (144%) compared to the maximum $NO_x$ increase seen in Fig. 11 l (66%), due to reducing effects by $HRs(RO_2)$ for both NO oxidation and PAN formation.

We provided the additional explanation for our manuscript at L543-569.

**Referee's comment: 2.** Other than the well-known heterogenous processed of N2O5, HO2 and RO2 analyzed in this study, the heterogeneous renoxification process of HNO3 to reproduce NOx has previously been suggested in order to explain the model overestimate of HNO3/NOx ratio in the free and polluted atmosphere (Hauglustaine et al., Geophys. Res. Lett., 23, 2609-2612, 1996: Lary et al., J. Geophys. Res., 102, 3671–3682, 1997; Li et al., SOLA, 11, 124–128, 2015; Akimoto et al., Atmos. Chem. Phys., 19, 603-615, 2019). Although the importance of this process has not been established, the same tendency of overestimate

of HNO3 and underestimate of NOx has been revealed in this study (Table 5). Discussion should be given for the possibility of the heterogeneous reaction of HNO3 whether in supporting or objecting.

**Author's response:** We thank for the suggestion from the Referee. In the original text, the comparison with ground observations for EANET and EMEP still showed low correlations for $HNO_3$ (0.177 for EANET, 0.116 for EMEP – Table 5). In Fig. S9, the model correlations with EANET and EMEP observations for $HNO_3$ and $NO_x$ showed higher tendencies for $HNO_3$ overestimates and $NO_x$ underestimates at low levels of $HNO_3$ (0-1 ppb) and high levels of $NO_x$ (>10 ppb), which indicate the highly polluted sites. For remote regions covered by ATom1 flights, our model showed relatively large overestimates for $NO_x$ at the surface layer (Fig. 6 a,e). The low reproducibility of model for $NO_x$ could be due to the low horizontal resolution of the simulations (~ 2.8°). Higher resolutions could improve the model reproduction for surface $NO_x$ as previously investigated by Sekiya et al. (Geosci. Model Dev., 11, 959–988, 2018).

The heterogeneous "renoxification" reaction of $HNO_3$ on soot surface (R4), which is suggested by the Referee, could also be a possible solution: $HNO_3 + soot \rightarrow NO + NO_2$ (R4).
The additional (R4) followed by $NO_2$ uptakes onto soot: $NO_2 + particles \rightarrow 0.5 HONO + 0.5 HNO_3$ (R5), can be expected to increase NO, and decrease $O_3$ via the consequent titration reaction. These changes could reduce the model overestimates for $HNO_3$ and $O_3$, and the model underestimates for $NO_x$ with EANET and EMEP stations.

A concerning $NO_x$ chemistry regarding HONO formation is already considered in another report (preparing for submission), coupling several HONO reactions including (R5). Without (R4)'s inclusion, the whole HONO chemistry could either increase or decrease $HNO_3$ at EMEP and EANET stations during winter and summer conditions, resulting in slight reductions for model bias with EANET and EMEP for $HNO_3$. However, the comparison with ground observations for $NO_x$ was not improved. When we incorporate (R4) into the model, the $NO_x$ chemistry did not undergo an effective "renoxification" to enhance $NO_x$ concentrations over EANET. To be able to conclude whether the "renoxification" process could remedy the issue, further examination would be required.

We revised our manuscript based on the above explanation at L267-276.

**Referee's comment: 3.** Many of the figures are rather poorly presented for readers and should be revised. (1) In most of the figures, size of inside letters and axis labels are too small (unreadable on print and difficult to read even on PC screen). (2) Fig.3: How the site for each species were selected? There is no explanation in the text. (3) Figs. 3, 4, 5: The difference between the plots for noHR_n2o5, _ho2, _ro2 and _CLD are almost undiscernible. It is suggested to show only noHR and STD in these Figures, and the difference of noHR_n2o5, _ho2, _ro2 and _CLD should be presented in some selected plots in a different Figure. (4) Figs. 9, 11, 12: The differences between the upper and lower figures are not discernible easily. It is suggested to delete the figures for HRS(N2O5-aerosols), HRS(HO2-Cloud) and HRS(RO2-Cloud) in these Figures. It would be enough to explain in the text that the uptake of N2O5 on aerosols, and that of HO2 and RO2 on cloud are major processes. Explanation should be given in the text why the process predominate for each of the species. (5) Figs. 10, 14: Labels and units of horizontal axis should be given properly.

**Author's response:** (1), (5) We acknowledged the responsibility for the figures' readability. We considerably modify each figure in the revised version upon the Referee's suggestions.

(2) We revised Fig. 3 with presenting the median value of grouped stations as Chinese region (stations in China and South Korea), remote stations with low $NO_x$ levels of EANET, and all EMEP stations.

(3) In Fig. 3,4,5, we separate plots of each HR impacts from the concentration plots, at which we keep only noHR and STD's comparison with measurements.

(4) Figures for HRS(N2O5-aerosols), HRS(HO2-Cloud) and HRS(RO2-Cloud) are moved to the Supplement.

**Referee's comment: 4.** Table 2: What is the meaning of asterisk for "product*". What do the ISO2 and MACRO2 stand for?

**Author's response:** The asterisk for "product*" in Table 2 was meant to represents a remaining error of expression and was deleted in the revised version. ISO2 denoted for peroxy radicals from $C_5H_8$+OH, and MACRO2 stands for peroxy radicals from the oxidation of MACR, methacrolein ($CH_2=C(CH_3)CHO$). These descriptions were provided in the revised version at L194-195.

**Referee's comment: 5**. Tables 5, 6, 7, 8: Units should be given appropriately.

**Author's response:** We provided the units in these tables appropriately in the revised version.

Sincerely,

On behalf of all co-authors,

Phuc T. M. Ha.

Dear the Anonymous Referee #2,

We appreciate your time and effort dedicated to providing valuable feedbacks on our manuscript.

**Referee's comment:** This manuscript presents a global model-based study of the impacts of various heterogeneous uptake processes on tropospheric composition. The CHASER model is validated against a number of ground-, ship-, aircraft-, and satellite-based observations of relevant species (NOx, PM, O3, CO, OH, and total column O3) and parameters (cloud fraction). A variety of simulations are designed to probe aerosol, cloud droplet, and ice particle uptake of N2O5, HO2, and RO2, with individual species and uptake pathways (cloud versus aerosol) turned off in turn. Total and spatially-/temporally-resolved changes in methane lifetime, NOx, O3, and CO are assessed. Finally, a sensitivity simulation is conducted to evaluate the impacts to atmospheric composition with variations in the examined heterogeneous loss rates.

The study presents a well-rounded analysis of heterogeneous uptake from a global model perspective. The model is thoroughly assessed against a reasonable number of available observations. The thoroughness of sensitivity simulations, both turning on/off all and individual heterogeneous uptake reactions as well as varying the magnitude of the first-order loss rate applied, addresses in a methodical way how this chemistry might impact global composition. I think there may be a missed opportunity in this manuscript to more thoroughly discuss the mechanisms through which these impacts manifest, but diagnostics necessary to perform such assessments may be lacking, and this should not preclude publication of this work. There is also a concern that model biases (such as the overestimation in cloud fraction in the northern Pacific lower troposphere) will introduce model-dependent errors in the results, though I regard the acknowledgment of this issue in the text as sufficient. I would consider this work as suitable for publication in this journal following the incorporation of the suggestions noted below, primarily concerned with clarifications and organization.

**Author's response:** We are grateful to Reviewer # 2 for the insightful and possitive comments on our manuscripts. We thank you for recognizing some deficiencies of the current manuscript relates to model-dependent errors, e.g. overestimation of the model CHASER against the low troposphere cloud for the North Pacific region, and the opportunity for in-depth discussions of the mechanisms involved in the effects of heterogeneous reactions. We also thank Reviewer #2 for understanding our acceptable coverage of these shortcomings. We have been able to incorporate changes to reflect most of the suggestions provided by the Reviewer #2.

**Referee's comment: Major comment 1.** In addition to collective improvements to figure clarity and organization, noted below under minor comments, my only other major request would be to expand on discussions of the mechanisms underlying some of the changes exhibited in the presented results. For instance, the description of effects on PAN production and transport, ~L492, could be more explicit and re-emphasized in the Conclusions.

The reasons for the large increases in NOx near the surface in the Arctic during JJA due to HR(RO2) are still unclear to me – if there were a large source of NOx here, the reduction in PAN formation may make sense, but large sources at these high latitudes seems unlikely.

Similarly, why are there increases in NOx during DJF due to HR(RO2) in the high latitude southern oceans, just offshore?

**Author's response:** L543-L548: For the Arctic ocean during JJA, large increases in $NO_x$ near the surface are due to the reduction of PAN caused by $HR(RO_2)$, as described in the current manuscript, in association with the suppress for NO oxidation via (R11) which is described in the revised version. These two reasons resulted in the double increase for $NO_x$ by $HR(RO_2)$ (144%) as compared to that by $HR(HO_2)$ (66%).

(R11) $RO_2 + NO \rightarrow RO + NO_2$

The $NO_x$ increases in DJF at high latitudes of southern oceans' offshore could also relate to reduced transport of $NO_x$ due to reduced PAN formation, since these offshores are in the downwind areas of major BVOCs sources from South America, South Africa, and Australia. Moreover, the areas with significant $NO_x$ increases in Fig. 12 (right panels) are all linked with high-cloud SAD (Fig. S14 left panels). The additional discussion was added accordingly.

L670-672: We also added discussions on $HR(RO_2)$ effects in the conclusion, regarding its effects to PAN and $NO_x$ transportations and the reducing effect to CO.

**Referee's comment: Major comment** 2. Similarly, one impact attributed to RO2 uptake is a decrease in CO (e.g., L257). I'm curious about the mechanism, and not aware of any discussion regarding this. I would assume that the CO decrease is due simply to reductions in secondary production – functionalized C-containing RO2 species that would otherwise be oxidized to form CO are instead taken up on aerosols/cloud droplets. Do the authors know if this is the case, or if there is another mechanism at play? I understand that diagnosing these kinds of questions from global model output, especially from lengthy and numerous simulations, may be difficult, given limitations on how much output can be generated. While model evidence to further describe these mechanistic questions would be ideal, hypotheses supported from prior literature or simpler, logical arguments would suffice.

**Author's response:** L561-564: We agree that there are lack of discusion on the CO's decreasing impact due to $HR(RO_2)$, which differs from the increasing impacts due to $HR(N_2O_5)$ and $HR(HO_2)$. As adviced by the Referee, CO decrease might be due to reduction in CO's secondary production from oxidation of functionalized $RO_2$ species ($RO_2 \rightarrow$ HCHO/RCHO or ROOH $\rightarrow$ CO) such as isoprene (Kelvin and Jacob, Atmos. Chem. Phys., 19, 9613–9640, 2019) when these $RO_2$ species undergo instead the heterogeneous reactions on aerosols and clouds particles.

**Minor comments**:
**Referee's comment:** L120: Please add a reference for the MAC reanalysis biomass burning emissions, or else provide more detail

**Author's response:** L121: Reference for the MACC reanalysis system was added (Inness et al., 2013).

**Referee's comment:** ~L175: Somewhere in Section 2, the timeframe of the simulations should be clearly stated. I gather from some of the time series figures that model output is available for at least 2010-2018 – were all sensitivity simulations run for this entire period?

**Author's response:** L180: All the standard and sensitivity runs were conducted in 2009-2017 timeframe, with using 2009 forthe spin-up year. We added a sentence regarding the simulation timeframe.

**Referee's comment:** ~L255: The discussion surrounding Fig. 3 refers to differences between the various model sensitivity runs, but it is very difficult to make out the different colored lines representing the different simulations in the figure. Perhaps an inset that shows a "zoomed in" view of a representative portion of each panel, or else plotting in different coordinates, like % difference compared to obs versus time, would help remedy the issue. This applies to Figures 4 and 5 as well.

**Author's response:** L285: We removed the colored lines representing sensitivity runs and keep only observation (grey), noHR (black) and STD simulations (red) in Fig. 3, to better focus on the overall improvement of the model with HRs inclusion. Additional plots for heterogeneous effects on $NO_x$, $O_3$, CO caused by each HR were added to Fig. 3 to support the discussion on the different HR effects. Similar modifications were applied to Fig. 4 and 5 as well. Fig. 1, Fig. 6, Fig. 9 to 14, Fig. S15, S16 were also changed for better visualization.

**Referee's comment:** L286: The suggestion of insufficient downward mixing of stratospheric air in the model while CO is underestimated by the model seems counterintuitive to me. Stratospheric air should be depleted in CO, so I would expect higher observed CO would point to something other than stratospheric influence, especially at the surface, as the shipbased observations are. I'm unfamiliar with Kanaya et al.; do they provide some other rationale to explain this apparent discrepancy?

**Author's response:** L318-322: We agree that the suggestion of insufficient downward mixing of stratospheric air in the model could only explain the underestimates by the model for $O_3$ in NP and Artic regions (Fig. 4 b: T1, T4, T5, T6). Model's underestimation for CO in the same region ($< 30$ ppbv) should be explained by insufficient emissions for CO as we used the HTAP-II inventory, as the CO biases are only minor in Kanaya et al. (2019) which used reanalysis data by inverse modeling as emission input to CHASER. In the revised manuscript, we modified the reason of CO's underestimation by model as insufficient emissions for CO.

**Referee's comment:** L334: It is unclear to me why the "ground layer" is defined differently for all flights ($> 800$ hPa) versus for the N. Pacific region ($> 700$ hPa) in Table 8 – could the authors include a brief explanation?

**Author's comments:** L376-377: We agree that the given sentence was unclear. With the base idea is that "HR($HO_2$) seems only to reduce the model bias in a thin layer: from the ground up to 800 hPa for all flights and 700 hPa for the North Pacific region ", the sentence will be modified as above.

**Referee's comment:** L365: I feel that the TCO plots in Fig. 7 would be more easily understood by plotting Model – OMI differences, for both the STD and noHR simulations. As is, the differences between the model runs and OMI stand out far more than the differences between the STD and noHR runs. One really has to focus on small details to see where the model is improving with respect to OMI.

**Author's comments:** ~L420: Thank you for the suggestion. We exchanged the original Fig. 7 with the plots of STD – OMI and noHR – OMI differences for TCO.

**Referee's comment:** L466: The statement that "recent O3 increases can be attributed to reduced HO2 uptake under aerosol (PM) decreases brought about by the new Chinese Air Pollution policy," is, I think, too strongly worded without a quantitative accounting

of the observed O3 changes. Other effects, such as the non-linearity in O3 production with NOx concentration, could also be contributing. Qualifying the statement as "can be attributed in part to reduced HO2. . ." or similar, would be sufficient.

**Author's response:** L514: The text was modified as " the observed recent $O_3$ increases can be attributed in part to reduced HO2 uptake under aerosol (PM) decreases brought about by the new Chinese Air Pollution policy.".

**Referee's comment:** L548: "magnitude of HRs" is vague; could you clarify if this sensitivity test is meant to probe uncertainties in the first-order loss rate, possible non-linearities in the uptake, etc.?

**Author's response:** L620: The sensitivity test is meant to test the effective-oxidation sensitivity of the troposphere in case future pollution and climate change might enhance the activities of these HRs, e.g. enhance the surface aerosol density Aj in Eq. (8). In other word, this test is not meant to probe the uncertainties in the first-order loss rate, but meant to probe the possible non-linearities in the response of tropospheric oxidation capacity to the linear enhancement of the loss rate, due to the complexation of tropospheric chemistry. Thus we modified the phrase "magnitude of HRs" to "magnitude of loss rate".

**Referee's comment:** L589: I'd suggest staying consistent with the number of significant figures reported in the % changes here, in the abstract (L13), and elsewhere. Sometimes one digit is reported after the decimal place, sometimes two, and sometimes none.

**Author's response:** L14: We have modified the figure followed by % change, with two decimal places for global average changes, and no decimal places for changes at the regional level (e.g. North region Pacific or China).

**Referee's comment:** Figs. 10 and 14: x-axis labels would be helpful, for anyone who may miss the (%) in the title.

**Author's response:** x-axis labels with unit % were added into Figs. 10 and 14 (Fig. 14 was updated as Fig. 13).

**Referee's comment: technical corrections**.
L130: "uncertainties" should be "uncertain" → L131
L293: "undervalues" is a slightly out-of-place word choice; "underestimates" may be better → L328
L297: The use of "extends" here suggests that model underestimates are extending in time/space instead of getting worse. "worsens" or "exaggerates" may better reflect the intended meaning. → L332
L301: A verb is needed in this sentence; "Ocean is mostly dominated. . ." → L336
L325: "However, for the. . ." the "for" is not needed. → L368
L326: Here and elsewhere, I'd suggest the authors check for consistency in how "ATom1" is capitalized and punctuated. →L369
L364: The order of appearance of the Supplemental figures should match the order in which they are mentioned in the main text. → L410
L380: "surface aerosol density" should be "surface area density" → L426
L415: "preferably onto" confuses the meaning of this sentence; I suggest "rather than onto" to emphasize the importance of aerosol uptake over cloud uptake → L461
L503: "glob" should be "globe" → L573
L502: Fig. 14 is introduced here before Fig. 13 is discussed; I'd suggest switching the two. → L572

**Author's response:** All modifications are made as suggested, highlighted in the according lines.

**Referee's comment:** L536: The phrase "N2O5 uptake on aerosols are mostly ascribed" would be more easily understood as "N2O5 uptake is mostly ascribed to aerosols"

**Author's response:** L606: The meaning is not only "$N_2O_5$ uptake is mostly ascribed to aerosols" in the mid and upper troposphere, but also the aerosols $N_2O_5$ uptake is the most dominant HRs in these atmospheric layers. So we change it to "the $N_2O_5$ uptake on aerosols is dominant in these layers".

Sincerely,

On behalf of all co-authors,
Phuc T. M. Ha.